# Cellular profiling of a recently-evolved social behavior in cichlid fishes

Zachary V. Johnson [1,2,3,4,5] ✉, Brianna E. Hegarty[1,2,5], George W. Gruenhagen[1,2,5], Tucker J. Lancaster[1,2], Patrick T. McGrath[1,2] ✉ & Jeffrey T. Streelman [1,2] ✉

Social behaviors are diverse in nature, but it is unclear how conserved genes, brain regions, and cell populations generate this diversity. Here we investigate bower-building, a recently-evolved social behavior in cichlid fishes. We use single nucleus RNA-sequencing in 38 individuals to show signatures of recent behavior in specific neuronal populations, and building-associated rebalancing of neuronal proportions in the putative homolog of the hippocampal formation. Using comparative genomics across 27 species, we trace bower-associated genome evolution to a subpopulation of glia lining the dorsal telencephalon. We show evidence that building-associated neural activity and a departure from quiescence in this glial subpopulation together regulate hippocampal-like neuronal rebalancing. Our work links behavior-associated genomic variation to specific brain cell types and their functions, and suggests a social behavior has evolved through changes in glia.

Social behaviors vary within and among species, and they are disrupted in heritable human brain diseases[1,2]. Much progress in understanding the biological mechanisms of social behaviors has been made through work in diverse and non-traditional model systems, in part because many social behaviors are not expressed in traditional laboratory models[3]. These advances have been made through largely independent experimental traditions spanning genomics[4], endocrinology[5], and circuit neuroscience[6,7]. However, we still have a poor understanding of how variation in the genome changes specific cell populations in the brain to generate variation in social behavior.

In this study, we investigate the neural and genomic substrates of bower-building behavior, a recently-evolved (estimated <500 ka) social behavior in Lake Malawi cichlid (*Cichlidae*) fishes[8]. Cichlids are a large group of behaviorally and eco-morphologically diverse teleost fishes that share homologous genes and brain cell populations with other vertebrates, including cell populations in the telencephalon that regulate social behaviors across vertebrate lineages[9,10]. Within Lake Malawi, cichlids have radiated into ~800 behaviorally diverse[11–13] but genetically similar species[8,14]. In ~200 species, males express

bower construction behaviors during the breeding season. During bower construction, males repetitively manipulate sand with their mouths, ultimately giving rise to a species-specific geometric structure. These structures serve as social territories that males aggressively defend against intruders, as well as mating sites for courtship and spawning with females[15]. Many species dig crater-like "pits" while others build elevated "castles" Pit-digging versus castle-building behavioral differences are associated with genomic divergence in a ~19 Mbp chromosomal region enriched for human disease-associated genes and genes that exhibit *cis*-regulated behavior-associated expression in the brain[16].

Here we integrate single nucleus RNA-sequencing (snRNA-seq), comparative genomics, spatial transcriptomics, and automated behavior analysis to systematically profile the telencephalon during castle-building behavior in *Mchenga conophoros*. We use natural genetic variation to link single nuclei back to 38 paired behaving/control test subjects, enabling analysis of building-associated signals while controlling for additional biological variables that vary among individuals, such as quivering, a courtship "dance" behavior, and relative gonadal mass. We map the cellular diversity of the telencephalon and profile

[1]School of Biological Sciences, Georgia Institute of Technology, Atlanta, GA 30332, USA. [2]Institute of Bioengineering and Bioscience, Georgia Institute of Technology, Atlanta, GA 30332, USA. [3]Department of Psychiatry and Behavioral Sciences, Emory University, Atlanta, GA, USA. [4]Emory National Primate Research Center, Emory University, Atlanta, GA, USA. [5]These authors contributed equally: Zachary V. Johnson, Brianna E. Hegarty, George W. Gruenhagen. ✉e-mail: zvjohns@emory.edu; patrick.mcgrath@biology.gatech.edu; todd.streelman@biology.gatech.edu

signatures of neuronal excitation, neurogenesis, and glial function, as well as genomic signatures of behavioral evolution across cell populations. Our work shows how snRNA-seq profiling can link natural behavior-associated genome variation to specific brain cell populations and their behavior-associated functions in uncharted species.

## Results

### Castle-building is associated with increased quivering behavior and gonadal physiology

Bowers (Fig. 1A, B) are constructed intermittently over many days. We used an automated behavior analysis system to monitor reproductive adult *Mchenga conophoros* males as they freely interacted with four reproductive adult females and sand (Fig. 1C). The system uses depth sensing to measure structural sand change and action recognition to identify building and quivering behaviors from video data[15,17]. We dissected telencephala (Fig. 1D) simultaneously from pairs of males in which one male was actively castle-building ($n = 19$) and the other was not (control, $n = 19$; Fig. 1E, F), and analyzed building and quivering behavior over the 100-min period preceding collection. We also tracked the gonadal somatic index (GSI), a measure of relative gonadal mass that is correlated with gonadal steroid hormone levels and social behaviors in cichlids[18,19] (Supplementary Data File 1). The volume of structural change was positively correlated with the number of

building events predicted from video data ($t_{36} = 10.78$, $R = 0.87$, $p = 8.15 \times 10^{-13}$; Fig. S1), and we combined these measures into a single Bower Activity Index (BAI). Building males had greater BAIs (Fig. 1G; $t_{18} = 9.02$, $p = 4.24 \times 10^{-8}$, two-tailed paired $t$-test), quivered more (Fig. 1H; $t_{18} = 6.10$, $p = 9.18 \times 10^{-6}$), and had greater GSIs (Fig. 1I; $t_{18} = 2.72$, $p = 0.0142$) compared to controls, but quivering and GSI were not predicted to mediate or moderate BAI (Supplementary Results). These results are consistent with castle-building, like many social behaviors in nature, being embedded within a suite of physiological and behavioral changes linked to reproduction.

### Telencephalic nuclei reflect major neuronal and non-neuronal cell classes

Dissected telencephala ($n = 38$) were combined into ten pools ($n = 5$ behave, $n = 5$ control, 3–4 telencephala/pool) for snRNA-seq (Fig. S2). >3 billion RNA reads were sequenced and aligned to the Lake Malawi cichlid *Maylandia zebra* reference genome[20]. 33,674 nuclei passed quality control filters and were matched to test subjects using genomic DNA. Clustering grouped nuclei into 15 primary (1°) and 53 secondary (2°) clusters (ranging from 57–1905 nuclei, Fig. 2A). Cluster composition was balanced across individuals (Fig. S3 and Supplementary Data File 3), and established marker genes revealed the presence of expected neuronal and non-neuronal cell types (Fig. 2B–D), including

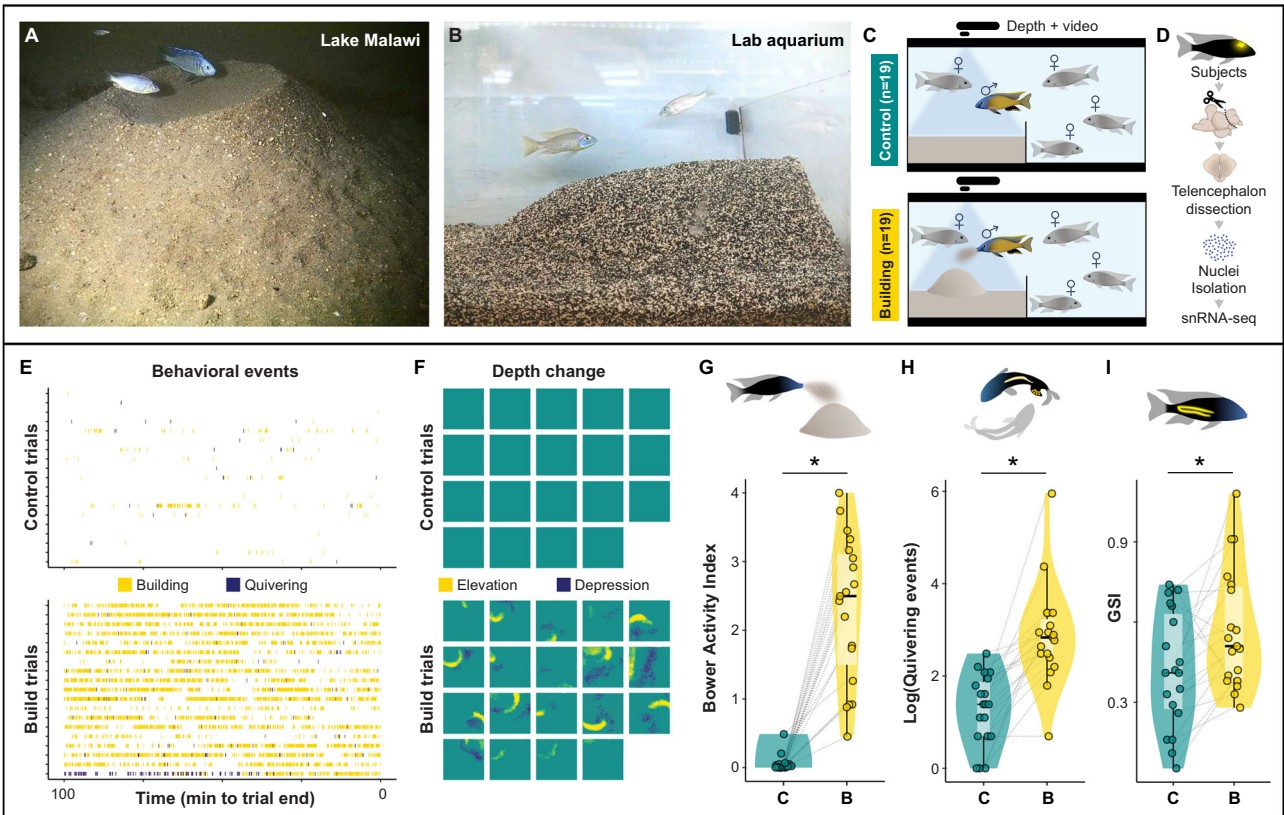

**Fig. 1 | Behavioral paradigm. A** *Mchenga conophoros* male (blue head) and female (silver) above a castle bower in Lake Malawi (photo credit Ad Konings). **B** *Mchenga conophoros* male (blue head) and female (silver) above a castle bower in a laboratory aquarium. **C** Schematic of the behavioral assay, 19 pairs of building (bottom) and control (top) males were sampled. **D** Simplified schematic of wetlab pipeline for snRNA-seq. **E** Action recognition (each trial is represented by a row; each tick mark indicates a behavioral event predicted by action recognition; paired males are matched by row at top and bottom) and **F** depth sensing (each square represents total depth change for one trial, with pairs matched by row and column between top and bottom panels) show behavioral differences between building and control males. Compared to controls, building males exhibited greater **G** BAIs, **H** quivering

behaviors, and **I** GSIs (gray lines link paired building/control males); $n = 38$ biologically independent animals ($n = 19$ building males, $n = 19$ control males). In all box plots, the center line indicates the median, the bounds of the box indicate the upper and lower quartiles, and the whiskers indicate 1.5x interquartile range. Asterisks indicate significance at $\alpha = 0.05$. Source data are provided as a Source Data file, and additional related data can be found in Supplementary Data 1. Fish artwork in panels **C**, **D**, **G**, **I** is reprinted from iScience, Vol 23 / Issue 10, Lijiang Long, Zachary V. Johnson, Junyu Li, Tucker J. Lancaster, Vineeth Aljapur, Jeffrey T. Streelman, Patrick T. McGrath, Automatic Classification of Cichlid Behaviors Using 3D Convolutional Residual Networks, 2020, with permission from Elsevier.

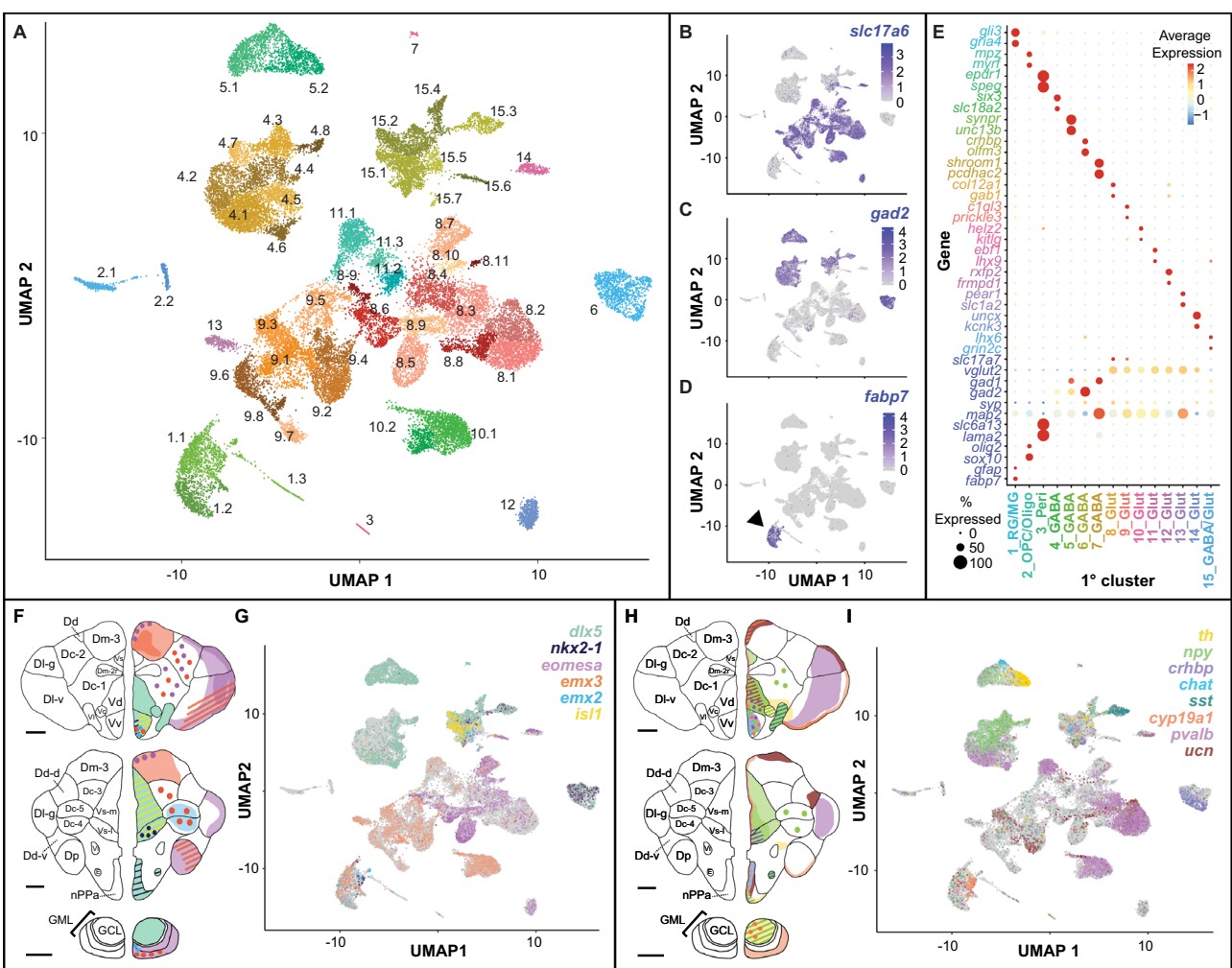

**Fig. 2 | Cellular diversity of the cichlid telencephalon. A** Nuclei cluster into 1° (*n* = 15) and 2° (*n* = 53) clusters. **B–D** Known marker genes reveal distinct clusters of **B** excitatory neurons (*slc17a6*+), **C** inhibitory neurons (*gad2*+), and **D** radial glia (*fabp7*+), as well as other less abundant cell types (see Supplementary Data File 2). **E** Clusters are distinguished by genes exhibiting nearly cluster-exclusive expression (top rows) as well as established cell type marker genes (bottom rows). **F–I** Genes encoding conserved **F, G** nTFs and **H, I** ligands (or ligand-related proteins) that exhibit conserved neuroanatomical expression profiles in teleost fishes (schematic representations of conserved expression patterns shown in panels **F, H** show distinct cluster-specific expression profiles). Anatomical figures adapted with permission from Dr. Karen Maruska. Data related to this figure can be found in Supplementary Data 2, 3.

excitatory neurons (*map2*+, *slc17a6*+), inhibitory neurons (*map2*+, *gad2*+), radial glia (*gfap*+, *fabp7*+), oligodendrocytes (*olig2*+, *mpz*+), oligodendrocyte precursor cells (OPCs, *olig2*+, *olig1*+), microglia (e.g., *apoeb*+, *mrc1*+), pericytes (*rgs5a*+, *pdgfrb*+), and hematopoietic stem cells (*runx1*+, *spi1*+). Unbiased analyses identified genes exhibiting nearly cluster-exclusive expression (Fig. 2E, top rows; Fig. S4). Genes encoding transcription factors (TFs) and neuromodulatory signaling molecules that show conserved brain region-specific expression patterns in teleosts were also preferentially expressed in distinct clusters (examples in Fig. 2F–I and Supplementary Data File 2). Marker genes for each 1° and 2° cluster were independently enriched (*q* < 0.05) for eight GO categories related to cell morphology, connectivity, conductance, and signal transduction (Supplementary Data File 4), supporting these as central axes distinguishing clusters in this study. Cluster marker genes were more strongly enriched for genes encoding conserved neurodevelopment/neuroanatomy-associated TFs (nTFs, *n* = 43) and ligands (*n* = 35) compared to neuromodulatory receptors (*n* = 108, Supplementary Data File 5; Fig. S5; and Supplementary Results), supporting more labile expression of receptor genes across cell types and consistent with recent single-cell RNA-seq (scRNA-seq) analyses of the mouse hypothalamus[21].

**Building, quivering, and gonadal physiology are associated with signatures of neuronal excitation in distinct cell populations**

To identify cell populations that may regulate castle-building behavior, we first investigated transcriptional signatures of neuronal excitation. Neuronal excitation induces transcription of conserved immediate early genes (IEGs) that typically peak in expression ~60–90 min later[22]. IEGs have thus become widely-used tools for identifying neuronal populations excited in response to specific stimuli or behavioral contexts[23]. However, IEG transcripts are recovered at relatively low levels in sc/snRNA-seq data[24]. To better track IEG signals, we identified genes that were selectively co-expressed with each of three established IEGs (*c-fos*, *egr1*, *npas4*) independently across 2° clusters (see Methods). In total, we identified 25 IEG-like genes (Supplementary Data File 6), most of which were known IEGs (*n* = 17), but eight of which had not been previously described (predicted homologs of human *DNAJB5*, *ADGRB1*, *GPR12*, *ITM2C*, *IRS2*, *RTN4RL2*, and *RRAD*; Fig. 3A). We assigned each nucleus an "IEG score" equal to the total number of IEG-like genes expressed. To disentangle building-associated signals from quivering- and GSI-associated signals, we fit a sequence of models in which BAI over the previous 100 min (Fig. 1G), quivering over the previous 100 min (Fig. 1H), and GSI (Fig. 1I) competed in different combinations

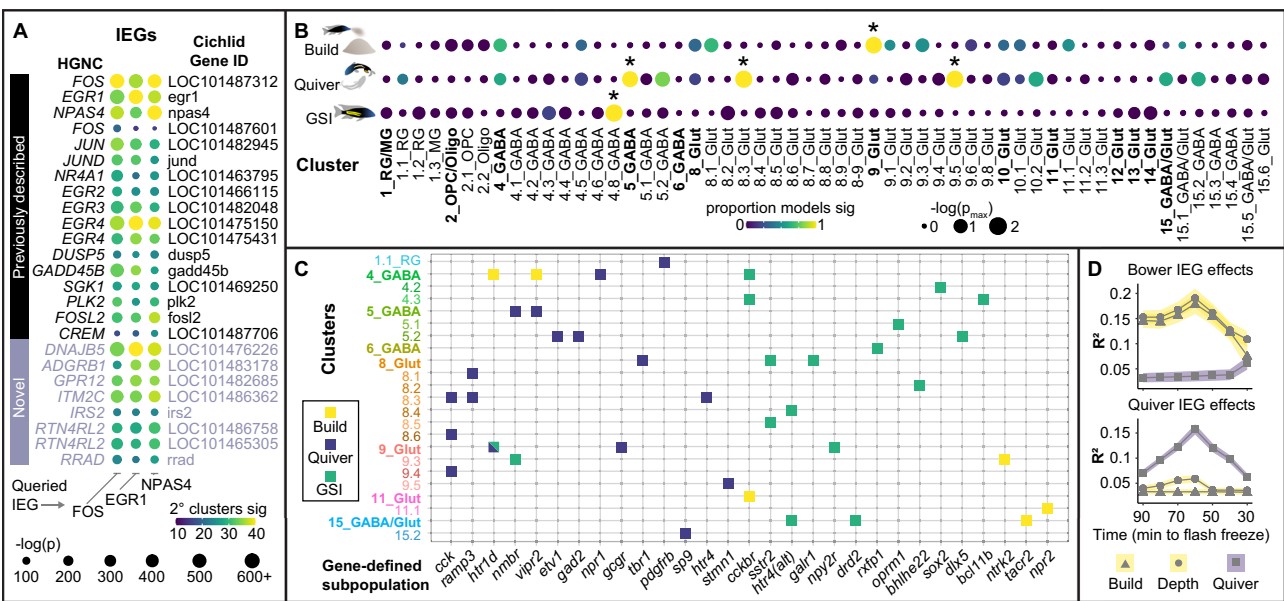

**Fig. 3 | Distinct cell populations exhibit building-, quivering-, and gonadal-associated IEG expression. A** Twenty-five genes were selectively co-expressed with *c-fos*, *egr1*, and *npas4* across cell populations; genes that were detected in the majority of clusters, and that were significantly ($p < 0.05$, Wilcoxon rank-sum test, two-sided) upregulated in IEG-positive nuclei compared to IEG-negative nuclei in the majority of those clusters were considered significant. **B** Building-, quivering-, and gonadal-associated IEG expression was observed in distinct clusters (linear mixed-effects regression, two-sided, adjusted for 5% false discovery rate), and **C** gene-defined populations (filled squares indicate significant effects, linear mixed-effects regression, two-sided, adjusted for 5% false discovery rate; 9_Glut *htr1d*+ nuclei showed both quivering- and GSI-associated IEG expression). **D** IEG expression was most strongly associated with the amount of temporally binned building

(top, 30 min bins, $n = 22$ neuronal populations exhibiting significant or trending building-associated IEG expression) and quivering (bottom, 30 min bins, $n = 71$ neuronal populations exhibiting significant or trending quiver-associated IEG expression) behavior performed approximately 60 min prior to tissue freezing (peak bins represent behavior performed 45–75 min prior to freezing the telencephalon). Data were presented as mean values ± SEM (represented by colored bands). Source data are provided as a Source Data file, and additional related data can be found in Supplementary Data 6. Fish artwork in panel **B** is reprinted from iScience, Vol 23 / Issue 10, Lijiang Long, Zachary V. Johnson, Junyu Li, Tucker J. Lancaster, Vineeth Aljapur, Jeffrey T. Streelman, Patrick T. McGrath, Automatic Classification of Cichlid Behaviors Using 3D Convolutional Residual Networks, 2020, with permission from Elsevier.

to explain IEG score across clusters (see Methods). We considered effects significant if the raw $p$ value was <0.05 in all models, and if the FDR-adjusted harmonic mean $p$ value across models was significant ($hmp_{adj} < 0.05$)[25].

We identified distinct sets of cell populations that showed building-, quivering-, or GSI- associated IEG induction (Fig. 3B and Supplementary Data File 6). Building was associated with increased IEG score in 9_Glut ("build-IEG+", the effect of building versus control, $\beta_{build} = 0.11 \pm 0.046$; $hmp_{adj} = 0.0016$), a cluster exhibiting gene expression patterns reflective of the dorsal pallium (Supplementary Data File 2), but not in any other 1° or 2° cluster. We also reasoned that some behaviorally-relevant cell populations may not align 1:1 with clusters. Some may span multiple clusters—for example, neuropeptides can diffuse to modulate distributed cell populations expressing their target receptors[1]—or may be best described as distinct functional subsets of cells within a cluster. Therefore, we extended our analysis to populations defined by nTF, ligand, and receptor genes, as well as a small set of additional genes of interest ($n = 17$, "Other", Supplementary Data File 5), both within clusters and regardless of cluster assignment (see Methods). This revealed a suite of additional build-IEG + populations, including three populations defined regardless of cluster (*elavl4*+, *cckbr*+, *ntrk2*+), and 4_GABA *htr1d*+, 4_GABA *vipr2*+, 15_GABA/Glut *tacr2*+, 11_Glut *cckbr*+, and 11.1_Glut *npr2*+ nuclei (Fig. 3C and Supplementary Data File 6), consistent with a role for these molecular systems in the neural coordination of building. In contrast, quivering-associated IEG signals suggested the involvement of dopaminergic olfactory populations (Supplementary Results). As further reinforcement of the behavioral relevance of these signals, we temporally binned behavioral measures and found that both building- and quivering-associated IEG signals were most strongly associated with

behavior expressed ~60 min before flash freezing of the telencephalon (Fig. 3D), consistent with known nuclear RNA time courses[26]. Follow-up mediation analyses identified five build-IEG+ populations as potential mediators of BAI, including 9_Glut, 4_GABA *htr1d*+, 4_GABA *vipr2*+, 11.1_Glut *npr2*+, and *ntrk2*+ (Supplementary Results).

## Excitatory neuronal populations drive building-associated gene expression

We also used unbiased analyses of behavior-associated differentially expressed genes (DEGs) to identify candidate cell populations underlying castle-building. Indeed, social behaviors have been linked to large changes in brain gene expression in diverse lineages[16,27,28], but the roles of specific cell populations in driving these effects are not well understood. We found that DEGs associated with building (build-DEGs), quivering (quiver-DEGs), and GSI (gonad-DEGs) were over-represented in largely non-overlapping sets of neuronal clusters (Fig. 4A). Three excitatory neuronal clusters accounted for a disproportionate number of build-DEGs (Fig. 4A, top). A minority of neuronal clusters also accounted for disproportionate numbers of quiver-DEGs (Fig. 4A, middle) and gonad-DEGs (Fig. 4A, bottom; Supplementary Results, Supplementary Data File 7). Despite these non-overlapping patterns, a large set of build-DEGs, quiver-DEGs, and gonad-DEGs were the same genes ($n = 81$; Fig. 4B), consistent with behavior- and gonadal-associated recruitment of common transcriptional programs in distinct neuronal populations. Unexpectedly, build- and quiver-DEGs exhibited a strong direction bias, and were predominantly upregulated in both 1° and 2° clusters (Fig. 4C, four leftmost bars). In contrast, gonad-DEGs tended more modestly toward upregulation in 1° clusters and were not directionally biased in 2° clusters (Fig. 4C, two rightmost bars). Upregulated build-DEGs, quiver-

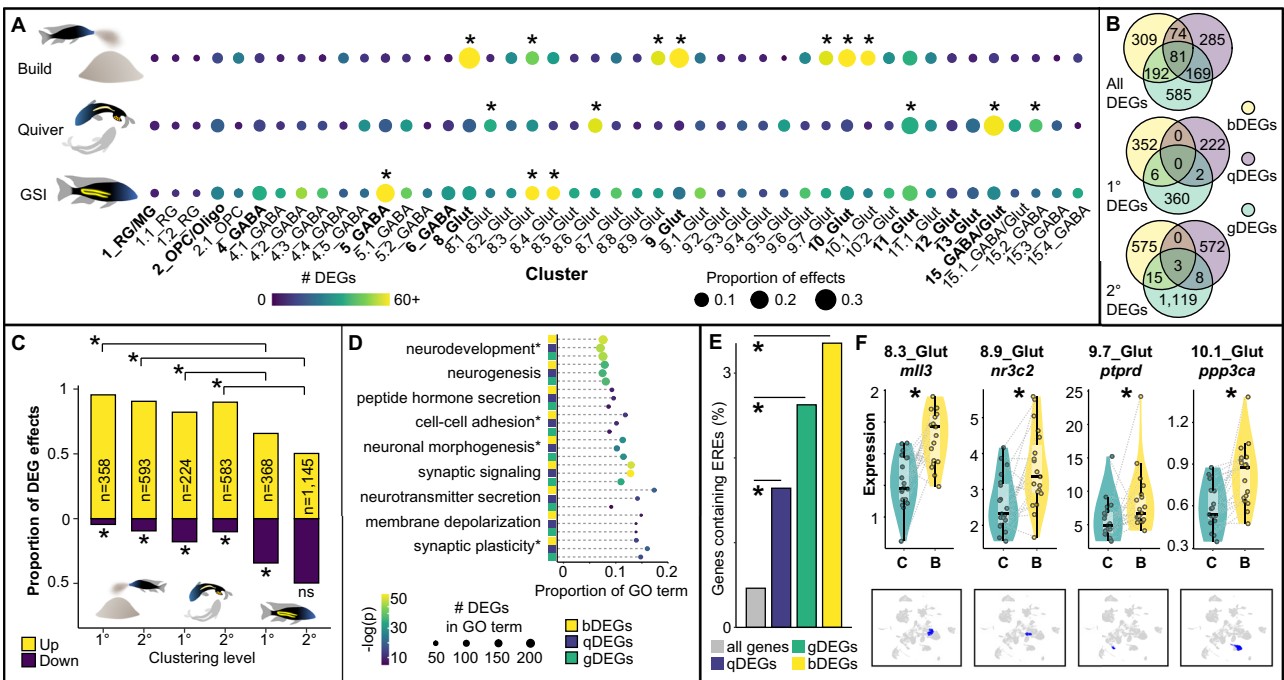

**Fig. 4 | Building, quivering, and GSI are associated with distinct patterns of cell type-specific gene expression. A** Build-DEGs (top row), quiver-DEGs (middle row), and gonad-DEGs (bottom row) were overrepresented in distinct sets of 1° and 2° clusters (FET, two-sided, asterisks indicate significance after adjustment for 5% false discovery rate). **B** Many of the same genes show building-, quivering-, and gonadal-associated expression ($n = 81$, top Venn diagram, overlap is considered regardless of the cluster in which the DEG effect was observed), but in distinct patterns across clusters (1° cluster DEGs, middle Venn diagram; 2° cluster DEGs, bottom Venn diagram; overlap considers the cluster in which the DEG effect was observed; low overlap among build-DEGs, yellow; quiver-DEGs, purple; gonad-DEGs, green). **C** Behavior-associated gene expression is driven by upregulation (first four bars; 1° build-DEGs, 342/358, $X^2(1, N = 358) = 184.96$, $p = 4.01 \times 10^{-42}$; 2° build-DEGs, 537/593, $X^2(1, N = 593) = 231.54$, $p = 2.75 \times 10^{-51}$; 1° quiver-DEGs, 184/224, $X^2(1, N = 224) = 50.19$, $p = 1.39 \times 10^{-12}$; 2° quiver-DEGs, 524/583, $X^2(1, N = 583)$, $p = 1.81 \times 10^{-49}$; Pearson's Chi-squared test with Yates' continuity correction, two-sided), whereas gonadal-associated gene expression is driven by a balance of up- and downregulation (last two bars; 1° gonad-DEGs, 242/368, $X^2(1, N = 368) = 18.1$, $p = 2.09 \times 10^{-5}$; 2° gonad-DEGs, 576/1,145, $X^2(1, N = 1145) = 0.011$, $p = 0.92$; Pearson's Chi-squared test with Yates' continuity correction, two-sided). **D** Build-DEGs, quiver-DEGs, and gonad-DEGs are enriched for many of the same GO terms related to synaptic structure, function, and plasticity; neurotransmission; and neurogenesis (Hypergeometric test, one-sided, Bonferroni corrected; GO terms followed by asterisks are abbreviated). **E** build-DEGs, quiver-DEGs, and gonad-DEGs are enriched for estrogen response elements compared to other genes throughout the genome (left barplot; build-DEGs, yellow, odds ratio = 7.33, $CI_{95} = [4.42, 11.60]$, $p = 7.26 \times 10^{-12}$; quiver-DEGs, purple, odds ratio = 3.53, $CI_{95} = [1.65, 6.72]$, $p = 9.23 \times 10^{-4}$; gonad-DEGs, green, odds ratio = 5.63, $CI_{95} = [3.57, 8.57]$, $p = 1.42 \times 10^{-13}$; FET, two-sided). **F** Violin plots show example effects for build-DEGs containing estrogen response elements in four different clusters (gray lines link paired building/control males, blue labeling below violin plots highlights clusters in which each effect was observed); $n = 38$ biologically independent animals ($n = 19$ building males, $n = 19$ control males). In all box plots, the center line indicates the median, the bounds of the box indicate the upper and lower quartiles, and the whiskers indicate 1.5x interquartile range. Asterisks indicate significance after adjustment for a 5% false discovery rate. Source data are provided as a Source Data file, and additional related data can be found in Supplementary Data 7–9. Fish artwork in panels **A** and **C** is reprinted from iScience, Vol 23 / Issue 10, Lijiang Long, Zachary V. Johnson, Junyu Li, Tucker J. Lancaster, Vineeth Aljapur, Jeffrey T. Streelman, Patrick T. McGrath, Automatic Classification of Cichlid Behaviors Using 3D Convolutional Residual Networks, 2020, with permission from Elsevier.

DEGs, and gonad-DEGs were enriched ($q < 0.05$) for a large number of the same GO terms (468 Biological Processes, 132 Cellular Components, 107 Molecular Functions; Supplementary Data File 8), the strongest of which were related to synaptic transmission, synaptic plasticity, and neurogenesis/neuronal differentiation (Fig. 4D). These results are consistent with building-, quivering-, and gonadal-associated gene expression reflecting, in part, changes in synaptic structure/function and neuronal differentiation.

Based in part on these enrichment patterns, we directly investigated estrogen as a possible regulator of behavior- and gonadal-associated gene expression. Estrogen is a steroid hormone that regulates vertebrate social behaviors[29], neuronal excitability[30], neurogenesis[31], and gene expression (by binding to estrogen receptors and forming a TF complex that binds to estrogen response element sequences in DNA)[32]. Build-DEGs, quiver-DEGs, and gonad-DEGs were independently enriched for estrogen response elements (Fig. 4E, F; Supplementary Results; genes containing estrogen response elements in Supplementary Data File 9), consistent with a role for estrogen in regulating male behavior- and gonadal-associated gene expression. Build-DEGs that contained estrogen response elements ($n = 22$ genes) were most strongly enriched for

GO terms including "modulation of chemical synaptic transmission" (top GO Biological Process, $q = 2.30 \times 10^{-4}$) and "Schaffer collateral-CA1 synapse" (top Cellular Component, $q = 2.22 \times 10^{-5}$), consistent with a role for estrogen in shaping building-associated changes in synaptic function.

## Castle-building is associated with neuronal rebalancing in the putative homolog of the hippocampal formation

We next investigated building-associated differences in cluster proportions, reasoning that differences could be caused by building-associated neurogenesis (e.g., changing rates of new neuronal influx into specific populations), or by building-associated gene expression (changing how nuclei are assigned to clusters). We identified two 2° clusters showing building-associated changes in proportions: 8.4_Glut ($\beta_{build} = 0.37 \pm 0.11$, $q = 0.013$) and 8.1_Glut ($\beta_{build} = -0.44 \pm 0.10$, $q = 7.67 \times 10^{-4}$; Fig. 5A–C). These clusters were not immediate neighbors in UMAP space, and thus the pattern was not simply explained by building-associated shifts in gene expression. The relative proportions of 8.4_Glut versus 8.1_Glut were negatively correlated across subjects ($t_{1,36} = -3.51$, $R = -0.50$, $p = 0.0012$; Fig. S6), and mediation analyses

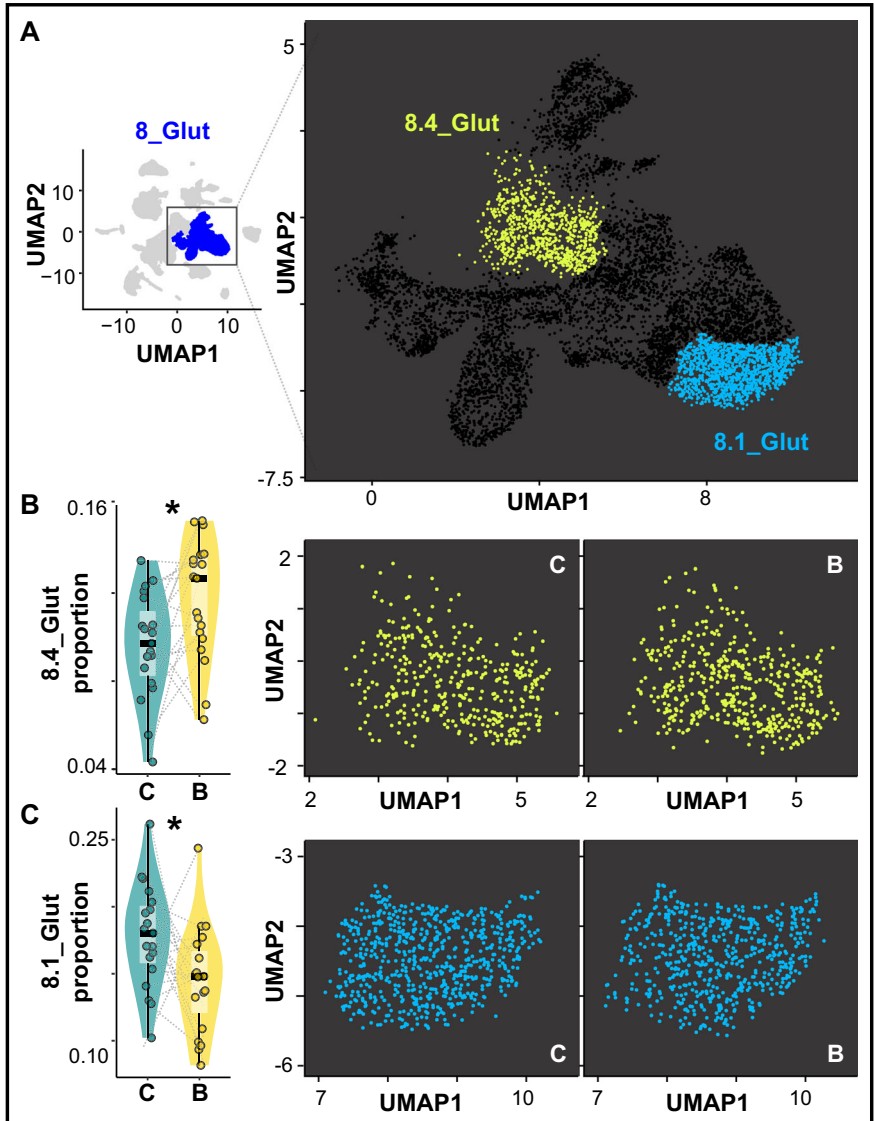

**Fig. 5 | Building-associated changes in the relative proportions of excitatory neuronal subpopulations. A** Two neuronal populations within 8_Glut exhibit building-associated changes in proportions. **B** 8.4_Glut exhibits a building-associated increase in its relative proportion within 8_Glut (two right panels show 8.4_Glut nuclei in control, "C," versus building "B," males, pooled across individuals; linear mixed-effects regression assuming a binomial distribution, two-sided, adjusted for 5% false discovery rate, $q = 0.013$), $n = 38$ biologically independent animals ($n = 19$ building males, $n = 19$ control males). **C** 8.1_Glut exhibits a building-associated decrease in its relative proportion within 8_Glut (two right panels show 8.1_Glut nuclei in control versus building males, pooled across individuals; linear mixed-effects regression assuming a binomial distribution, two-sided, adjusted for 5% false discovery rate, $q = 7.67 \times 10^{-4}$), $n = 38$ biologically independent animals ($n = 19$ building males, $n = 19$ control males). Gray lines link paired building/control males. In all box plots, the center line indicates the median, the bounds of the box indicate the upper and lower quartiles, and the whiskers indicate 1.5x interquartile range. Asterisks indicate significance at $\alpha = 0.05$ after adjusting for a 5% false discovery rate. Source data are provided as a Source Data file, and additional related data can be found in Supplementary Data 3.

supported these proportion changes as possible mediators of both BAI and build-IEG+ signals (including 9_Glut, 4_GABA *htr1d*+, 4_GABA *vipr2*+, and *ntrk2*+; Supplementary Results). The encompassing 1° cluster, 8_Glut, was distinguished by markers of the lateral region of the dorsal telencephalon (Dl; Supplementary Data File 2), a region that regulates spatial learning in other fish species[33] and that is putatively homologous to the mammalian hippocampal formation based on gene expression, cell morphology, connectivity, anatomy, and behavioral evidence[9,34,35]. Together these data suggest that building-associated reorganization of hippocampal-like cell populations may be important for building behavior.

To further investigate signatures of building-associated neurogenesis, we analyzed the expression of 87 genes with the GO annotation "positive regulation of neurogenesis" in both zebrafish and mice

("proneurogenic" genes, Supplementary Data File 10). Building, in contrast to quivering and GSI, was associated with widespread increases in proneurogenic gene expression across clusters (Fig. S7). Building-associated expression of proneurogenic genes was disproportionately strong in populations expressing estrogen receptors (Normalized Enrichment Score, NES = 2.00, $q = 0.034$; Fig. S7, Supplementary Results), consistent with a role for estrogen signaling in building-associated neurogenesis. Notably, 8.4_Glut showed building-associated proneurogenic gene expression ($\beta_{build} = 0.08 \pm 0.033$, $hmp_{adj} = 0.028$), and expression of proneurogenic genes in this cluster was positively correlated with its relative proportion ($t_{1,36} = 2.12$, $R = 0.33$, $p = 0.041$). These results support the idea that building-associated changes in neurogenesis underlie building-associated increases in 8.4_Glut proportion.

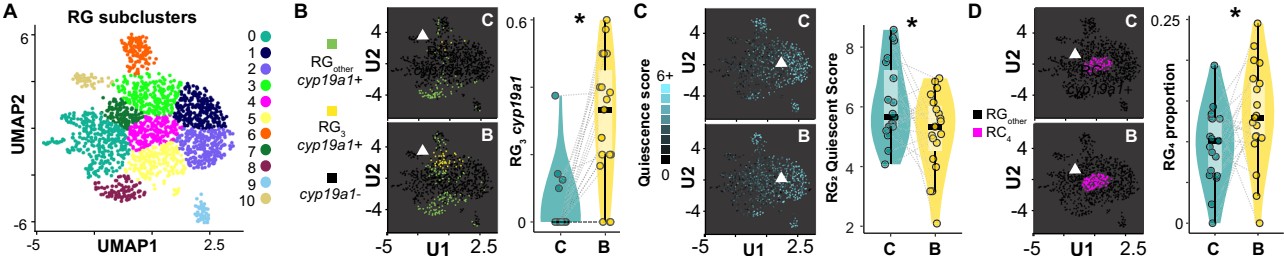

Fig. 6 | **Building-associated changes in radial glial biology. A** Re-clustered radial glial subpopulations show building-associated **B** *cyp19a1* expression (RG$_3$), **C** signatures of decreased quiescence (RG$_2$), and **D** increases in proportion (RG$_4$); $n = 38$ biologically independent animals ($n = 19$ building males, $n = 19$ control males). Gray lines link paired building/control males. UMAP plots in the right three panels represent effects in control ("C," top) and building ("B," bottom) males, pooled across individuals. In all box plots, the center line indicates the median, the bounds of the box indicate the upper and lower quartiles, and the whiskers indicate 1.5x interquartile range. Asterisks indicate significance at α = 0.05 after adjusting for a 5% false discovery rate. Source data are provided as a Source Data file, and additional related data can be found in Supplementary Data 7–9.

## Building behavior is associated with changes in glial cell biology

Radial glia are the primary source of new neurons in adult teleosts[36], and can occupy distinct functional states, including quiescence, cycling, and neuronal differentiation[37]. Therefore, we reasoned that if building is associated with changes in neurogenesis, then building might also be associated with changes in radial glial biology. To investigate building-associated changes in radial glial biology, we first analyzed building-associated gene expression within radial glia (1.1_RG and 1.2_RG pooled) and identified 25 build-DEGs that were collectively enriched for "neuron development" (top GO Biological Process, $q = 8.18 \times 10^{-4}$) and "astrocytic glutamate-glutamine uptake and metabolism" (top Pathway, $q = 0.0010$). Build-DEGs in radial glia included *cyp19a1* (upregulated; Fig. S7 and Supplementary Data File 10), the gene encoding aromatase, an enzyme that converts testosterone into brain-derived estrogen. Aromatase is thought to be exclusively expressed in radial glia in the teleost telencephalon, and has been linked to radial glial functional states as well as neurogenesis[38]. These data support building-associated changes in radial glial biology that may shape building-associated neurogenesis.

To investigate building-associated changes in radial glia at higher resolution, we re-clustered radial glia into 11 subclusters (RG$_0$-RG$_{10}$; Fig. 6A). Building-associated increases in *cyp19a1* expression were driven most strongly by RG$_3$ ($\beta_{build} = 1.44 \pm 0.42$, hmp$_{adj}$ = 0.018; Fig. 6B), a subpopulation distinguished by *lhx5* and *gli3*, both nTFs that regulate neurogenesis in mammals[39,40]. A second subpopulation, RG$_2$, exhibited ~3x more build-DEGs than any other subcluster (Supplementary Data File 11; 18/19 effects reflected building-associated downregulation). To further investigate radial glial subclusters, we assigned each radial glial nucleus a quiescence, cycling, and neuronal differentiation score based on marker genes for these functional states (Supplementary Data File 12). Building was associated with decreased quiescence score in RG$_2$ ($\beta_{build} = 0.29 \pm 0.11$, hmp$_{adj}$ = 0.010; Fig. 6C), but was not associated with quiescent, cycling, or neuronal differentiation score in any other subcluster. Lastly, we reasoned that building-associated changes in radial glial gene expression and/or functional states may result in changes in radial glial subcluster proportions. Analysis of subcluster proportions revealed an increase in the relative proportion of RG$_4$ ($\beta_{build} = 0.73 \pm 0.19$, $q = 0.0017$; Fig. 6D), which was positioned in UMAP space between subclusters expressing markers of quiescence (RG$_1$, RG$_2$) and those expressing markers of cycling and differentiation (RG$_0$; Supplementary Data File 13). Follow-up mediation analyses supported RG$_3$ *cyp19a1* expression as a possible mediator of building, and RG$_2$ quiescence, RG$_3$ *cyp19a1* expression, and RG$_4$ proportion as possible mediators of rebalancing (Supplementary Results). Together these data demonstrate building-associated changes in specific subpopulations of radial glia, and

highlight RG$_2$, RG$_3$, and RG$_4$ as candidate regulators of building-associated neurogenesis and rebalancing.

## Genes that diverged in castle-building species show behavior- and gonadal-associated upregulation

We next investigated if genome divergence associated with behavioral evolution may reveal additional candidate cell populations underlying building. The evolution of bower-building behavior has previously been linked to a ~19 Mbp region on Linkage Group 11 (LG11), within which genetic variants have diverged between closely-related castle-building and pit-digging lineages[16]. We performed follow-up comparative genomics analyses across 27 total Lake Malawi species (Supplementary Data File 14) and identified 165/756 genes in this region that additionally showed signatures of divergence between castle-building lineages and more distantly related rock-dwelling species that do not build bowers ("castle-divergent" genes, CDGs; Fig. 7A; Supplementary Data File 15). Thus, CDGs bear strong genomic signatures of castle-building evolution across Lake Malawi species. CDGs were expressed at higher levels in the telencephalon compared to neighboring genes in the same 19 Mbp region and compared to other genes throughout the genome (Fig. 7B). CDGs were also overrepresented among 1° and 2° cluster markers (Fig. 7C, D), and among upregulated build-DEGs (Fig. 7E), quiver-DEGs (Fig. 7F), and gonad-DEGs (Fig. 7G). These data support the behavioral significance of CDGs in the telencephalon, and suggest that castle-building has evolved in part through variation in genes that are upregulated during reproductive contexts.

## A subpopulation of pallial glia links genome evolution to hippocampal-like neuronal rebalancing

We next tested if CDGs were enriched in specific cell populations. CDGs were most strongly enriched in non-neuronal clusters (1.1_RG, 1.2_RG, 2.1_OPC), followed by neuronal clusters (4.3_GABA and 4.4_GABA) and gene-defined cell populations (5.2_GABA *th*+, and 9_Glut *hrh3*+; Supplementary Results, Fig. S8 and Supplementary Data File 16). We hypothesized that these enrichment patterns were driven by subsets of CDGs that were co-expressed together in specific cell populations. Indeed, a weighted gene correlation network analysis (WGCNA)[41] across nuclei revealed a module of 12 CDGs ("CDG module") that were strongly co-expressed compared to other CDGs (stronger correlation coefficients, Welch *t*-test, $p = 8.83 \times 10^{-14}$; stronger silhouette widths, Welch *t*-test, $p = 0.016$; Fig. 8A). Across clusters, the CDG module was most strongly enriched in 1.2_RG (compared to all other nuclei; Cohen's $d = 4.22$, CI$_{95}$ = [4.14,4.30], $p_{perm} = 0$), followed by 1.1_RG (Cohen's $d = 2.86$, CI$_{95}$ = [2.79,2.92], $p_{perm} = 0$; Fig. 8B and Supplementary Data File 17), suggesting both enrichment in radial glia as well as differences in expression among radial glial subpopulations.

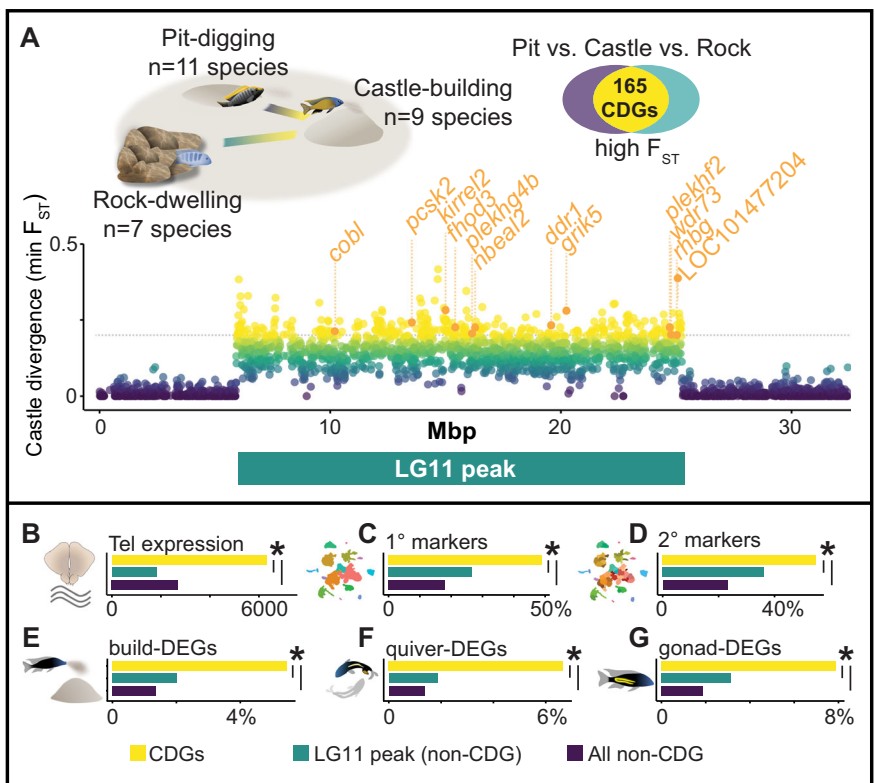

**Fig. 7 | Castle-divergent genes show distinct and building-associated expression patterns in the telencephalon. A** Genome comparisons of 27 species that build castles ($n = 9$), dig pits ($n = 11$), or do not construct bowers ($n = 7$) reveal 165 CDGs (CDG module genes labeled in orange). **B** CDGs (yellow) are expressed at higher levels in the telencephalon compared to other genes within the 19 Mbp region on LG11 (turquoise, $p_{perm} = 1.42 \times 10^{-5}$, two-sample permutation test, two-sided) and compared to other genes throughout the genome (purple, $p_{perm} = 1.77 \times 10^{-6}$, two-sample permutation test, two-sided; x-axis represents summed normalized expression values across all nuclei, averaged for each category). **C** CDGs are overrepresented among 1° cluster markers compared to other genes within the 19 Mbp region on LG11 (odds ratio = 3.61, $CI_{95} = [2.42,5.38]$, $p = 8.31 \times 10^{-11}$, FET, two-sided) and compared to other genes throughout the genome (odds ratio = 3.12, $CI_{95} = [2.25,4.31]$, $p = 1.43 \times 10^{-11}$, FET, two-sided). **D** CDGs are overrepresented among 2° cluster markers compared to other genes within the 19 Mbp region on LG11 (odds ratio = 3.08, $CI_{95} = [2.12,4.50]$, $p = 1.66 \times 10^{-9}$, FET, two-sided) and compared to other genes throughout the genome (odds ratio = 3.13, $CI_{95} = [2.27,4.30]$, $p = 1.51 \times 10^{-12}$, FET, two-sided). **E** CDGs are overrepresented among build-DEGs compared to other genes within the 19 Mbp region on LG11

(odds ratio = 4.19, $CI_{95} = [1.41,12.72]$, $p = 0.0044$, FET, two-sided) and compared to other genes throughout the genome (odds ratio = 2.82, $CI_{95} = [1.26,5.53]$, $p = 0.0066$, FET, two-sided). **F** CDGs are overrepresented among quiver-DEGs compared to other genes within the 19 Mbp region on LG11 (odds ratio = 5.19, $CI_{95} = [1.86,15.14]$, $p = 5.79 \times 10^{-4}$, FET, two-sided) and compared to other genes throughout the genome (odds ratio = 3.79, $CI_{95} = [1.84,7.02]$, $p = 3.01 \times 10^{-4}$, FET, two-sided). **G** CDGs are overrepresented among gonad-DEGs compared to other genes within the 19 Mbp region on LG11 (odds ratio = 4.50, $CI_{95} = [1.82,11.33]$, $p = 4.23 \times 10^{-4}$, FET, two-sided) and compared to other genes throughout the genome (odds ratio = 2.63, $CI_{95} = [1.37,4.67]$, $p = 0.0024$, FET, two-sided). Source data are provided as a Source Data file, and additional related data can be found in Supplementary Data 14, 15. Fish artwork in panels **A**, **E**, **G** is reprinted from iScience, Vol 23 / Issue 10, Lijiang Long, Zachary V. Johnson, Junyu Li, Tucker J. Lancaster, Vineeth Aljapur, Jeffrey T. Streelman, Patrick T. McGrath, Automatic Classification of Cichlid Behaviors Using 3D Convolutional Residual Networks, 2020, with permission from Elsevier. Drawings of pit-digging and rock-dwelling cichlids were adapted from original artwork published previously[15,74].

Within radial glia, CDG module score was positively correlated with quiescent score (Fig. 8C–E; $R = 0.61$, $CI_{95} = [0.58,0.63]$, $p = 8.65 \times 10^{-191}$); CDG module score was negatively correlated with cycling score ($R = -0.11$, $CI_{95} = [-0.15,-0.06]$, $p = 3.14 \times 10^{-6}$); and CDG module score was not correlated with neuronal differentiation score ($R = -0.036$, $CI_{95} = [-0.08,0.0090]$, $p = 0.12$). Analysis of TFs ($n = 999$) additionally revealed *npas3* as the most strongly co-expressed TF with the CDG module (Fig. 8F and Supplementary Data File 18). *npas3* suppresses proliferation in human glioma, is strongly expressed in quiescent neural stem cells, and is downregulated during hippocampal neurogenesis in mice[42,43]. Among radial glial subclusters, the CDG module was selectively enriched in RG$_1$ (compared to other radial glia; Cohen's $d = 0.68$, $CI_{95} = [0.55,0.82]$, $p_{perm} = 0.0196$) and RG$_2$ (Cohen's $d = 0.32$, $CI_{95} = [0.19,0.45]$, $p_{perm} = 0.046$; Fig. 8G and Supplementary Data File 17), both of which also strongly expressed markers of glial quiescence. Together these data support that the CDG module may be related to radial glial quiescence.

Building was associated with a decrease in CDG module score in RG$_2$ ($\beta_{build} = -0.26 \pm 0.11$, $hmp_{adj} = 0.027$; Fig. 8H), and an increase in CDG module score in RG$_8$ ($\beta_{build} = 0.61 \pm 0.24$, $hmp_{adj} = 0.0097$). The only CDG module gene exhibiting strict building-associated expression was *wdr73*, which was downregulated in RG$_1$ ($\beta_{build} = -0.08 \pm 0.0044$, $hmp_{adj} = 4.54 \times 10^{-89}$) and RG$_2$ ($\beta_{build} = 0.71 \pm 0.0070$, $hmp_{adj} = 2.47 \times 10^{-90}$; RG$_2$ effect in Fig. 8I). Notably, one study in human epithelial cells found that suppressed *wdr73* expression was most strongly associated with increased expression of *ccnd1*[44], a marker of neural stem cell proliferation[45]. Indeed, within radial glia, *wdr73* expression was negatively correlated with *ccnd1* expression ($R = -0.07$, 95% CI, $CI_{95} = [-0.12,-0.029]$, $p = 0.0013$). We hypothesized that a building-associated downregulation of the CDG module in RG$_2$ may promote an exit from quiescence and contribute to building-associated proportion changes in 8.1_Glut and 8.4_Glut. Consistent with this, the difference in relative proportion of 8.4_Glut and 8.1_Glut was explained by RG$_2$ CDG module score ($R = -0.53$, 95% CI, $CI_{95} = [-0.72,-0.25]$, $p = 6.71 \times 10^{-4}$;

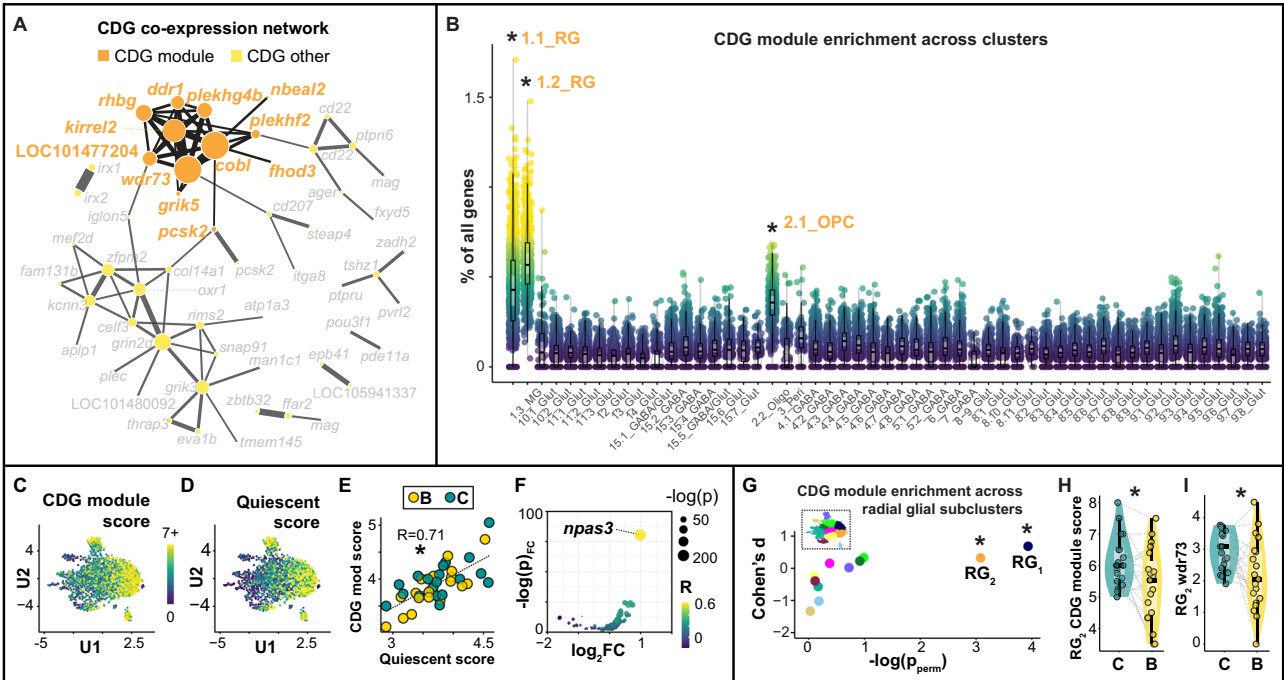

**Fig. 8 | A CDG module shows enriched and building-associated expression patterns in a subpopulation of radial glia. A** A weighted gene correlation network analysis identifies a module of 12 CDGs (orange dots connected by black lines) that are strongly co-expressed across telencephalic nuclei compared to other CDGs (yellow dots connected by dark gray lines). **B** The CDG module is most strongly enriched in radial glia (two leftmost columns), $n = 53$ biologically independent 2° clusters. Asterisks indicate effects that were significant after adjustment for a 5% false discovery rate and additionally as measured by a second permutation test. **C**, **D** CDG module expression across radial glia mirrors expression of quiescent markers. **E** Expression of the CDG module is positively correlated with the expression of radial glial quiescence markers. **F** *npas3* shows strong, positive,

outlier co-expression with the CDG module (Pearson's $R = 0.47$, $CI_{95} = [0.43, 0.50]$, $q = 3.19 \times 10^{-100}$). **G** Among radial glial subclusters, the CDG module is enriched in $RG_1$ and $RG_2$; $n = 11$ biologically independent radial glia subclusters. **H**, **I** $RG_2$ exhibits building-associated decreases in expression of **H** the CDG module and **I** *wdr73* in particular; $n = 38$ biologically independent animals ($n = 19$ building males, $n = 19$ control males). Gray lines link paired building/control males. In all box plots, the center line indicates the median, the bounds of the box indicate the upper and lower quartiles, and the whiskers indicate 1.5x interquartile range. Asterisks indicate effects that were significant after adjustment for a 5% false discovery rate. Source data are provided as a Source Data file, and additional related data can be found in Supplementary Data 15–18.

Fig. S9), *wdr73* expression ($R = -0.59$, $CI_{95} = [-0.76, -0.33]$, $p = 1.14 \times 10^{-4}$; Fig. S9), quiescent score ($R = -0.35$, $CI_{95} = [-0.76, -0.33]$, $p = 0.029$), and *npas3* expression ($R = -0.37$, $CI_{95} = [-0.62, -0.06]$, $p = 8.20 \times 10^{-4}$). Most of these relationships were evident within building males only (difference in 8.4_Glut and 8.1_Glut proportion versus $RG_2$ CDG module score, $R = -0.57$, $CI_{95} = [-0.81, -0.15]$, $p = 0.011$; versus $RG_2$ *wdr73* expression, $R = -0.62$, $CI_{95} = [-0.84, -0.23]$, $p = 0.0049$; versus $RG_2$ quiescent score, $R = -0.32$, $CI_{95} = [-0.68, 0.15]$, $p = 0.18$; versus *npas3* expression $R = -0.52$, $CI_{95} = [-0.79, -0.085]$, $p = 0.023$) but not within controls ($p \geq 0.24$ for all). In contrast, these relationships were not evident in $RG_1$ (across all subjects, $p \geq 0.17$ for all; within building males, $p \geq 0.12$ for all). These data are consistent with a role for CDG module expression in $RG_2$ in the rebalancing of 8.1_Glut and 8.4_Glut. In further support of this, mediation analyses identified both $RG_2$ *wdr73* expression and $RG_2$ CDG module score as candidate mediators of rebalancing (Supplementary Results).

In teleost fishes, anatomically distinct radial glial subpopulations supply new neurons to distinct brain regions (Fig. 9A). We used spatial transcriptomic profiling to investigate if the anatomical relationships among $RG_2$, 8.4_Glut, and 8.1_Glut were consistent with $RG_2$ supplying new neurons to 8.4_Glut and 8.1_Glut. 8.1_Glut mapped to ventral Dl-g, and 8.4_Glut mapped to ventral Dl-v (Fig. 9B–D), pallial subregions within Dl that receive new neurons from radial glia lining the pallial ventricular zone. $RG_2$ was anatomically positioned along the pallial but not subpallial ventricular zone (Fig. 9B–D), consistent with a potential to supply new neurons to Dl and other pallial regions. These data are further consistent with a relationship between $RG_2$ and neuronal rebalancing in 8.4_Glut and 8.1_Glut.

## Directional interaction among multiple neuronal and glial populations explain building

In the mammalian hippocampus, the activity of incoming axonal projections and local circuits is thought to regulate the differentiation of neural precursor cells into new neurons[46]. We hypothesized that building-associated neural activity may be causally related to building-associated changes in $RG_2$ and neuronal rebalancing. We first used CellChat to investigate the molecular potential for directional communication (cell–cell communication analysis) among build-IEG+ populations, $RG_2$, 8.1_Glut, and 8.4_Glut. For comparison, we also analyzed all other 1° and 2° clusters, radial glial subclusters, as well as randomized size-matched cell populations. Briefly, cell–cell communication analysis uses gene expression to estimate the molecular potential for communication between cell populations (measured in "connection weight" between a sender population and a receiver population) using known cell–cell adhesion and ligand-receptor binding proteins. Unlike most other tools, CellChat increases robustness by additionally accounting for heteromeric complexes and interaction mediator proteins[47]. CellChat revealed a strong molecular potential for directional communication from build-IEG+ populations to 8.4_Glut (Fig. S10; Supplementary Data File 19; Supplementary Results), consistent with neuronal populations that fire during building synapsing onto 8.4_Glut neurons. Because neuronal firing can strengthen synaptic connections, we next investigated if build-IEG+ populations also showed building-associated increases in connection weights with 8.4_Glut. Only ~2% (94/4,096) of all connection weights analyzed exhibited building-associated changes ($q < 0.05$, 91/94 were increases, Supplementary Data File 20), and these were enriched for

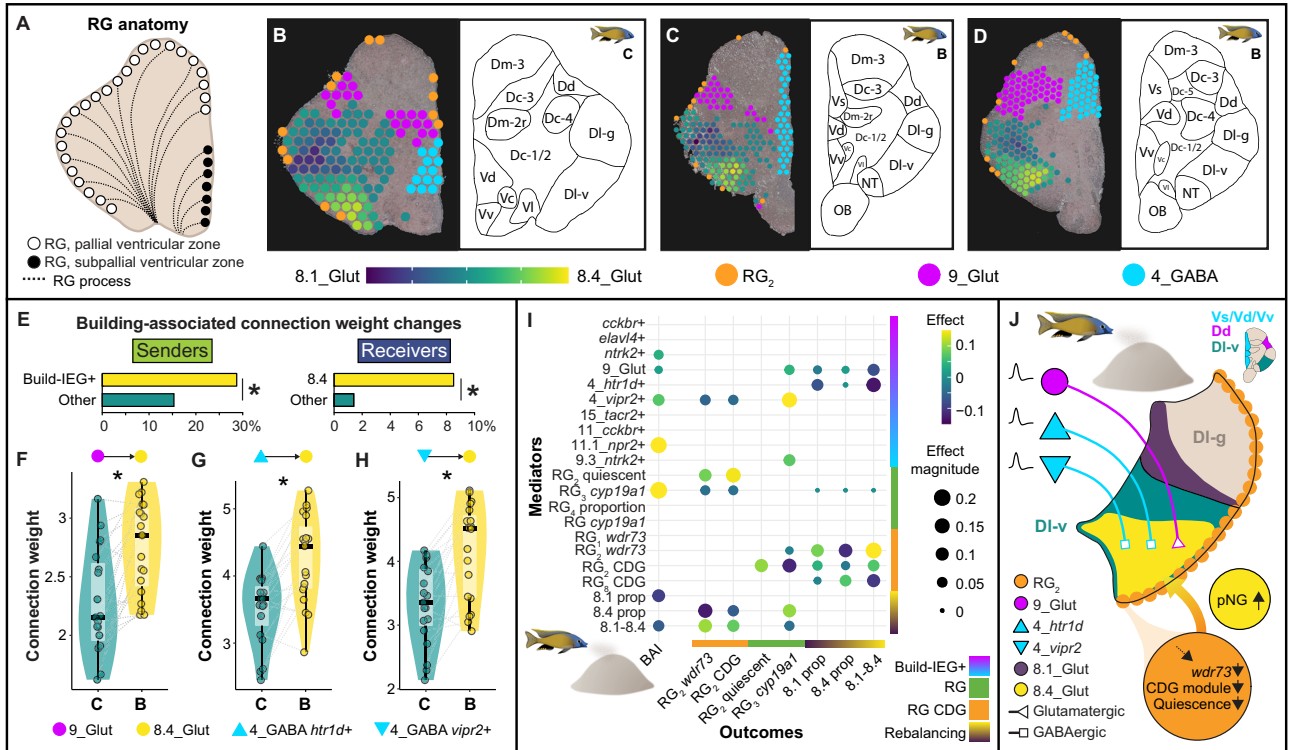

**Fig. 9 | A circuit model for bower behavior and cellular reorganization in Dl, the putative homolog of the hippocampal formation. A** Radial glia differs in morphology, function, and anatomical distribution (e.g., pallial versus subpallial ventricular zones). **B–D** RG2 (orange) aligns with the pallial but not subpallial ventricular zone, and 8.1_Glut versus 8.4_Glut aligns with ventral Dl-g versus ventral Dl-v, respectively. **E** Connections exhibiting building-associated increases in strength were enriched for build-IEG+ senders (yellow, left) and for 8.4_Glut as a receiver (yellow, right; x-axis reflects the percentage of connections exhibiting building-associated changes in weight), asterisks indicate significance at α = 0.05. **F–H** Building is associated with increased connection weights from senders **F** 9_Glut, **G** 4_GABA *htr1d*+, and **H** 4_GABA *vipr2*+ to 8.4_Glut (receiver), n = 38 biologically independent animals (n = 19 building males, n = 19 control males), asterisks indicate effects that were significant at α = 0.05 after adjusting for 5% false

discovery rate. In all box plots, the center line indicates the median, the bounds of the box indicate the upper and lower quartiles, and the whiskers indicate the 1.5x interquartile range. Gray lines link paired building/control males. **I** Regularized multiple mediation analyses identified top candidate regulators of behavior (left column) and supported directional interaction among build-IEG+ populations, radial glial subpopulations, and hippocampal-like neuronal rebalancing. **J** A hypothesized neuron-glia circuit model for castle-building behavior. Source data are provided as a Source Data file, and additional related data can be found in Supplementary Data 19–21. Fish artwork in panels **I, J** is reprinted from iScience, Vol 23/Issue 10, Lijiang Long, Zachary V. Johnson, Junyu Li, Tucker J. Lancaster, Vineeth Aljapur, Jeffrey T. Streelman, Patrick T. McGrath, Automatic Classification of Cichlid Behaviors Using 3D Convolutional Residual Networks, 2020, with permission from Elsevier.

build-IEG+ senders (27/94, odds ratio = 2.23, CI₉₅ = [1.36,3.56], p = 0.0013, Fisher's exact test, FET; Fig. 9E, left) and 8.4_Glut as a receiver (8/94, odds ratio = 6.55, CI₉₅ = [2.61,14.38], p = 8.78 × 10⁻⁵; Fig. 9E, right; and interestingly 2.1_OPC as a receiver, see Supplementary Results, Supplementary Data File 20). Specifically, four build-IEG+ senders showed increased connection weights with 8.4_Glut, including 9_Glut (β_build = 0.47 ± 0.13, hmp_adj = 0.0030; Fig. 9F), 4_GABA *htr1d*+ (β_build = 0.76 ± 0.25, hmp_adj = 0.0082; Fig. 9G), 4_GABA *vipr2*+ (β_build = 0.82 ± 0.23, hmp_adj = 0.0020; Fig. 9H), and *ntrk2*+ (β_build = 0.82 ± 0.28, hmp_adj = 0.012), all of which were previously identified as possible mediators of BAI (Supplementary Results). Spatial integration (see Methods) anatomically mapped 9_Glut to the dorsal region of the dorsal telencephalon (Dd) and 4_GABA to the dorsal/supracommissural regions of the ventral telencephalon (Vd/Vs; Fig. 9B–D), both of which send axonal projections to Dl-v in other teleosts[9,48].

Lastly, we performed follow-up mediation analyses to refine candidate directional relationships among build-IEG+ populations, radial glia, and neuronal rebalancing. These analyses converged on excitation in three build-IEG+ populations (9_Glut, 4_GABA *htr1d*+, and 4_GABA *vipr2*+) as well as aromatase expression in RG₃, as possible mediators of CDG module downregulation in RG₂ and neuronal rebalancing (Fig. 9I; Supplementary Results, Supplementary Data File 21). We identified two plausible ligand-receptor pathways positioned to facilitate

these interactions and support radial glial (RG₂) to neuronal (8.4_Glut) differentiation, including NRG2/NRG3 (putatively secreted from build-IEG+ populations) to ERBB4 (preferentially expressed in RG₂ and 8.4_Glut), and estrogen (putatively synthesized by aromatase in RG₃) to ESR2 (preferentially expressed in RG₂; see Supplementary Results, Fig. S11, S12). In summary, cell–cell communication, spatial transcriptomics, previous tracing experiments, mediation analyses, and gene expression together supported a testable circuit model in which projections from 9_Glut, 4_GABA *htr1d*+, 4_GABA *vipr2*+ to Dl-v regulate functional changes in subpopulations of radial glia, hippocampal-like neuronal rebalancing, and building behavior. The hypothesized model is that behavior-associated excitation in three neuronal populations, together with behavior-associated aromatase expression in RG₃, promotes the downregulation of the CDG module and a corresponding exit from quiescence in RG₂. This exit from quiescence promotes 8.1_Glut and 8.4_Glut neuronal rebalancing, a cellular reorganization that, in turn, facilities building behavior (Fig. 9J and Fig. S13).

## Discussion

Behaviors evolve through genomic variation and are executed through the activity and interactions of many heterogeneous cell populations. Historically, the inability to profile many cell populations simultaneously has constrained progress in identifying which populations coordinate specific behaviors and understanding how genetic

variation shapes their function. These goals have been advanced largely through the concerted efforts of many laboratories focusing on specific behaviors in the same model species over decades. Sn/scRNA-seq technologies disrupt this landscape by enabling (1) simultaneous functional profiling of many cell populations and (2) rapid tracing of behavior-associated genetic variation to specific cell populations in any species. We combined these approaches to show evidence that genome divergence associated with social behavioral evolution has altered a subpopulation of glia, changing neuron-glia interactions, downstream hippocampal-like cellular organization, and function of neural circuits underlying active castle-building behavior.

Our work builds on three proof-of-principle studies demonstrating the promise of sn/scRNA-seq technologies for profiling neuronal signatures of behavior[21,24,26]. However, these studies were conducted in C57BL6/J inbred mice (hindering matching of pooled cells to individual subjects), and functional profiling was focused on IEGs. Here we investigated a natural species and used individual genetic variation to match single nuclei to 38 test subjects, their recent behavioral histories, and their relative gonadal masses. This improved our power for analyzing behavior-associated IEG signals as well as behavior-associated changes in gene expression, cell type proportions, glial functional states, and predicted directional communication among cell populations. Converging evidence supported the involvement of three build-IEG+ populations (9_Glut, 4_GABA htr1d+, 4_GABA vipr2+) in building behavior. These populations showed building-associated expression of proneurogenic genes as well as signatures of directional and building-associated communication with 8.4_Glut, which in turn showed building-associated increases in relative proportion. Using spatial transcriptomics, we mapped these build-IEG+ populations to brain regions that send axonal projections to Dl-v (where 8.4_Glut mapped) in other teleost species. Notably, 9_Glut mapped to Dd, a dorsal pallial region that innervates Dl in a many-to-one fashion in other fish, mirroring a conserved hippocampal circuit that regulates pattern separation in mammals[49]. 9_Glut also accounted for a disproportionate number of build-DEGs and was identified as a candidate mediator of both building behavior and neuronal rebalancing. Thus, the combination of functional profiling, cell–cell communication, and spatial transcriptomic analyses converged on candidate cell populations, brain regions, and circuit projections underlying building.

Analyses of behavior-associated gene expression, proneurogenic gene expression, cluster proportion changes, radial glial functional states, and genome divergence also converged on an unanticipated role for neurogenesis in the evolution and expression of castle-building behavior. Brain region-specific cell proliferation and/or neurogenesis during species-specific social contexts occurs in diverse taxa[50,51]. In cichlids, changes in the social environment cause changes in cell proliferation within three hours, supporting relatively rapid behavior-associated changes in neurogenesis[52]. Multiple lines of evidence supported building-associated neurogenesis in 8.4_Glut, a cluster that mapped to a subregion of the putative hippocampal formation homolog that is associated with spatial learning in other fish[34]. In the wild, bowers are constructed through thousands of spatial sand manipulation decisions that give rise to a species-specific structure. It has been reported in several species that in response to damage or destruction during storms, males will repair or reconstruct their bowers to match their original size, geometry, and spatial location[53,54], suggesting spatial information is central to bower behavior. After the breeding season, bowers are abandoned. It is intriguing to speculate that building-associated neurogenesis in Dl-v is related to spatial representations of the bower that are maintained during the breeding season. Interestingly, seasonal changes in neurogenesis occur in songbirds that repeat their song within a breeding season but change their song between seasons. These birds show increased cell proliferation in vocal learning circuits during the breeding season that declines when the season is over[55,56].

Together our data led to a testable model linking behavioral and genome evolution to glia, hippocampal-like rebalancing, neuronal excitation, and active building behavior. We propose that castle-building behavior has evolved through divergence in a gene module that is selectively expressed in specific subpopulations of quiescent radial glia. In one subpopulation, the divergent gene module shows building-associated downregulation, and this downregulation corresponds to signatures of reduced glial quiescence and downstream cellular reorganization in the putative homolog of the hippocampus. We find further evidence that building-associated excitation in specific neuronal populations and aromatase expression in another glial subpopulation are important in coordinating these events. Notably, glia may represent a recurring cellular target for vertebrate brain and behavioral evolution, as genes regulated by human accelerated regions and gained enhancer regions are also enriched in this cell type.[57] Ultimately, our work provides an example of how specific genes, brain regions, and cell populations may have given rise to a new form of social behavior.

Interestingly, the CDG module resides in a 19 Mbp genomic region that exhibits signatures of chromosomal inversions[58–60]. Inversions can facilitate rapid evolution by protecting large-scale and adaptive cis-regulatory landscapes and multi-allele haplotypes (supergenes) from recombination[61,62]. Recent work in ruff and white-throated sparrows support roles for inversions in social behavioral evolution[63,64]. In our data, four CDG module genes, including wdr73, are immediately proximate to one end of the 19 Mbp region. It is intriguing to speculate that a divergent cis-regulatory architecture surrounding this putative inversion breakpoint has changed CDG regulation in glia to facilitate behavioral evolution.

## Methods
### Subjects
All cichlids (species *Mchenga conophoros*) used in this study were fertilized and raised into adulthood (>180 days) in the Engineered Biosystems Building cichlid aquaculture facilities at the Georgia Institute of Technology in Atlanta, GA, in accordance with the Institutional Animal Care and Use Committee guidelines (IACUC protocol number A100029). This colony was originally derived from wild-caught populations collected in Lake Malawi. All experimental animals were collected as fry at ~14 days post-fertilization from mouthbrooding females and were raised with broodmates on a ZebTec Active Blue Stand Alone system. At ~60 days post-fertilization, animals were transferred to 190-L (92 cm long × 46 cm wide × 42 cm tall) glass aquaria and were housed in social communities (20–30 mixed-sex individuals) into adulthood. Environmental conditions of aquaria were similar to those of the Lake Malawi environment; subjects were maintained on a 12-h:12-h light:dark cycle with full lights on between 8 a.m. and 6 p.m. Eastern Standard Time (EST) and dim lights on for 60 min between light-dark transition (7 a.m.–8 a.m. and 6 p.m.–7 p.m. EST) in pH = 8.2, 26.7 °C water and fed twice daily (Spirulina Flake; Pentair Aquatic Ecosystems, Apopka, FL, USA). All tanks were maintained on a central recirculating system. Reproductive adult subject males (age 6–14 months post-fertilization, $n = 38$) were visually identified from home tanks based on nuptial coloration and expression of classic courtship behaviors (i.e., chasing/leading, quivering). Reproductive adult stimulus females were visually identified from home tanks based on distension of the abdomen (caused by ovary growth) and/or buccal cavity (caused by mouthbrooding).

### Behavior tanks
Behavior tanks were equipped with LED strip lighting synced with external room lighting to minimize large shadows and maximize consistency in video data used for action recognition (10-h:14-h light:dark cycle). Sand (Sahara Sand, 00254, Carib Sea Inc.; ACS00222)

was contained within a 38.1 cm long × 45.6 cm wide section of each tank and separated from the rest of the aquarium by a custom 45.6 cm wide × 17.8 cm tall × 0.6 cm thick transparent acrylic barrier secured with plastic coated magnets (1.25 cm wide × 2.5 cm tall × 0.6 cm thick; BX084PC-BLK, K&J Magnetics, Inc.). This design ensured that all fish could freely enter and leave the enclosed sand tray region throughout the trial. At the start of the trial, the smoothed sand surface lay ~29.5 cm directly below a custom-designed transparent acrylic tank cover (38.1 cm long × 38.1 cm wide × 3.8 cm tall) that directly contacted the water surface to eliminate rippling for top-down depth sensing and video recordings.

## Behavior assays

Subject males were introduced to behavioral tanks containing sand and four reproductive adult age- and size-matched stimulus females of the same species. Broods were collected from all mouthbrooding females prior to the introduction of subject males to behavior tanks. Following the introduction, each male was allowed to acclimate to the novel behavior tank setup and to initiate castle-building to minimize potential confounding effects of novelty on brain gene expression that may be caused by introduction to a novel environment and/or a novel experience (i.e., the first experience building). After building was confirmed during this initial pre-trial and acclimation period, the sand surface in each behavioral tank was reset (i.e., the sand surface was smoothed) shortly before lights off, and behavioral trials were initiated in which male activity was recorded and remotely monitored over subsequent days using an automated depth sensing and video recording system as previously described in ref. 15. Briefly, the system uses a Raspberry Pi 3 mini-computer (Raspberry Pi Foundation), a Microsoft XBox Kinect Depth sensor to track the developing bower structure (snapshots captured every 5 min), and a Raspberry Pi v2 camera to record 10 h of high-definition video data daily. The system regularly uploads depth change updates to a Google Documents spreadsheet, enabling real-time, remote monitoring of bower construction activity for every male. Pairs of males (one building, one control) were collected on the same day at the same time (immediately back to back), as described in more detail in the "Tissue Sampling" section below. Following each trial, a trained 3D Residual Network[17] was used to predict male building and quivering behaviors from video data in the 100 min preceding collection, and depth data were analyzed as an additional measure of building behavior in the same period.

## Tissue sampling

Actively constructing males ($n = 19$) were identified through remote depth change updates and were collected between 11 a.m. and 2 p.m. EST (3–5 h after full lights-on and feeding, and >12 h after the most recent sand reset) to control for potential effects of circadian rhythm, experimenter activity, feeding, hunger, and anticipation of food on brain gene expression. At the same time, a neighboring male that was not constructing a bower (nor had initiated construction following reset) but could also freely interact with four females and sand, was also collected (control, $n = 19$). Immediately following collection, subjects were rapidly anesthetized with tricaine for rapid brain extraction, measured for standard length (SL, distance measured from snout to caudal peduncle) and body mass (BM), and rapidly decapitated for brain extraction. Telencephala (including olfactory bulbs) were dissected under a dissection microscope (Zeiss Stemi DV4 Stereo Microscope 8x - 32x, 000000-1018-455), in Hibernate AB Complete nutrient medium (HAB; with 2% B27 and 0.5 mM Glutamax; BrainBits) containing 0.2 U/µl RNase Inhibitor (Sigma). Immediately following dissection telencephala were rapidly frozen on powdered dry ice and stored at −80 °C. Testes were then surgically extracted and weighed to calculate gonadosomatic index (GSI = gonad mass/BM × 100) for each subject (subject information available in Supplementary Data File 1).

## Nuclei isolation

Nuclei were isolated following a protocol adapted from ref. 65 and optimized for cichlid telencephala. Immediately prior to single nuclei isolation, frozen telencephala were pooled into five biological replicates ($n = 3$–4 subjects/pool) per behavioral condition (building versus control). Pools were organized such that individuals within a pool had nearly identical telencephalic mass with the aim of equalizing the relative mass of tissue and the relative number of nuclei sampled from each subject within each pool. Additionally, paired constructing versus control pools were organized such that males in both pools were matched as closely as possible in relative age, body mass, and sampling dates. Frozen telencephalon tissue sample pools were transferred into chilled lysis buffer containing 10 mM Tris-HCL (Sigma), 10 mM NaCl (Sigma), 3 mM MgCl$_2$ (Sigma), 0.1% Nonidet P40 Substitute (Sigma), and Nuclease-free H$_2$O. The tissue was incubated on ice and lysed for 30 min with gentle rotation. Following lysis, 1.0 mL HAB medium was added and the tissue was rapidly triturated 20 rounds using silanized glass Pasteur pipettes (BrainBits) with a 500 µm internal diameter to complete tissue dissociation. Dissociated tissue were centrifuged (600×$g$, 5 min, 4 °C) and resuspended in 2.0 ml chilled wash and resuspension buffer containing 2% BSA (Sigma) and 0.2 U/µl RNase Inhibitor (Sigma, as described above "Tissue Collection") in 1X PBS (Thermo Fisher). The nuclei suspensions were filtered through 40 µm Flowmi® cell strainers (Sigma) and 30 µm MACS® SmartStrainers (Milltenyi) to remove large debris and aggregations of nuclei prior to fluorescence-activated cell sorting (FACS).

## Fluorescence-activated cell sorting

Pilot experiments revealed that multiplets (clumps of multiple nuclei adhered together) passed through both passive filtration steps, and therefore we further improved the quality and purity of our sample using FACS (BD FACSAria™ Fusion Cell Sorter, BD Biosciences) and FACSDiva software (v8.0.1, BD Biosciences). Sizing beads (6 µm; BD Biosciences) and 1.0 µg/ml DAPI (Sigma) were used to set gating parameters, enabling the selection of singlet nuclei based on size (forward scatter), shape (side scatter), and DNA content (DAPI fluorescence). Thus, this step efficiently filtered out multiplets and irregularly shaped nuclei (characteristic of unhealthy or dead nuclei). At least 300,000 nuclei/pool were collected into 1 mL wash and resuspension buffer for downstream sequencing. FACS collection data was visualized using FlowJo v10.6.0.

## snRNA-seq

Suspensions of isolated nuclei were loaded onto the 10x Genomics Chromium Controller (10x Genomics) at concentrations ranging from 400-500 nuclei/ul with a target range of 3,000–4,000 nuclei per sample. Downstream cDNA synthesis and library preparation using Single Cell 3′ GEM, Library and Gel Bead Kit v3.1, and Chromium i7 Multiplex Kit were performed according to manufacturer instructions (Chromium Single Cell 3′ Reagent Kits User Guide v3.1 Chemistry, 10X Genomics). Sample quality was assessed using high-sensitivity DNA analysis on the Bioanalyzer 2100 system (Agilent) and libraries were quantified using a Qubit 2.0 (Invitrogen). Barcoded cDNA libraries were pooled and sequenced on the NovaSeq 6000 platform (Illumina) on a single flow cell using the 300-cycle S4 Reagent kit (2x150 bp paired-end reads; Illumina).

## DNA sequencing

Genomic DNA was isolated from diencephalic tissue sampled from each test subject using a DNeasy Blood and Tissue Kit pipeline with a 60 min lysis time and without RNase A. The 260/280 nm absorbance ratio ranged from 1.91–2.10 across subjects. Libraries were prepared following a NEBNext Ultra II FS DNA Library Prep Kit for Illumina protocol. Libraries were sequenced on two NovaSeq 6000 lanes using 300-cycle SP Reagent Kits (2x150 bp paired-end reads; Illumina).

## Spatial transcriptomics

Telencephala were microdissected from two size-matched build-control pairs of *Mchenga conophoros* males (*n* = 4 males total), embedded in cryomolds, flash-frozen on dry ice, and stored at −80 °C until further processing. Tissue was cryo-sectioned coronally at 10-µm thickness at −20 °C (Cryostar NX70) and mounted onto pre-chilled Visium Spatial Gene Expression slides (10X Genomics). RNA quality (RIN >7) was confirmed on a Bioanalyzer 2100 system using an RNA 6000 Nano Kit (Agilent). Spatial gene expression slides were processed following manufacturer instructions (Visium Spatial Gene Expression Reagent Kits User Guide, 10X Genomics). Barcoded cDNA libraries were sequenced on the NovaSeq 6000 platform (Illumina) using the 150-cycle SP Reagent Kit.

## Quantification and statistical analysis

**Behavioral analysis.** For all trials, 3D ResNet-predicted behaviors and structural change across the sand surface was analyzed over the 100 min preceding collection following the same general approach described previously in ref. 17. Briefly, a smoothing algorithm was applied to remove depth change attributable to technical noise, and small regions of missing data were recovered by spatial interpolation. Bowers were defined as any region within which one thousand or more contiguous pixels (equivalent to ~10 cm²) changed in elevation by more than 0.2 cm in the same direction (~2 cm³ volume change total) based on previous analysis of depth change caused by non-building home tank activity[15]. Depth change values were adjusted based on the cubed standard length of each subject male, to create a standardized measure of building activity (larger males have larger mouths and can scoop and spit a larger volume of sand). Action recognition was used to track the number, location, and timepoints of predicted bower construction behaviors (scoops, spits, and multiple events) and quivering behaviors over the same 100 min period. The number of quivering events was log-normalized due to a single outlier (building) male with 257 predicted quivering events (~5.9 standard deviations above the mean). Feeding behaviors were not analyzed because they can be performed by both males and females and we are not able to reliably attribute individual feeding events to the subject male.

We generated a single Bower Activity Index (BAI) metric to reflect overall building activity by first calculating the regression line between depth change and building events for each trial (*n* = 38, $R^2$ = 0.76). We then projected each male's depth change and bower behavior values onto that line, with the lowest value (0 predicted building events, 0 above threshold depth change) being set to 0. BAI was calculated as the Euclidean distance along the regression line from the lowest value. BAI was used as a continuous measure of castle-building behavior throughout this study.

Differences in building, quivering, and GSI between groups were analyzed using a paired t-test in which behave and control subjects collected at the same time were treated as pairs.

## snRNA-seq pre-processing and quality control

FASTQ files were processed with Cell Ranger v3.1.0 (10X Genomics). Reads were aligned to the *Maylandia zebra* Lake Malawi cichlid genome assembly[20] using a splice-aware alignment algorithm (STAR) within Cell Ranger, and gene annotations were obtained from the same assembly (NCBI RefSeq assembly accession: GCF_000238955.4, M_zebra_UMD2a). Because nuclear RNA contains intronic sequences, introns were included in the cellranger count step. Cell Ranger filtered out UMIs that were homopolymers, contained N, or contained any base with a quality score of less than 10. Following these steps, Cell Ranger generated ten filtered feature-barcode matrices (one per pool) containing expression data for a total of 32,471 features (corresponding to annotated genes) and a total of 33,895 barcodes (corresponding to droplets and putative nuclei) that were used passed through additional quality control steps in the 'Seurat' package in R. Examination of

total transcripts, total genes, and proportion of mitochondrial transcripts were similar across all ten pools, and therefore the same criteria were used to remove potentially dead or dying nuclei from all pools. Barcodes associated with fewer than 300 total genes, fewer than 500 total transcripts, or greater than 5% (of total transcripts) of mitochondrial genes were excluded from downstream analysis on this basis. This step filtered out a total of 20 (0.059%) barcodes. To reduce the risk of doublets or multiplets, barcodes associated with more than 3000 total genes or 8000 total transcripts were also excluded. This step filtered out a total of 201 barcodes (0.59%). In total, 33,674 barcodes (99.34%) passed all quality control filters and were included in downstream analyses.

## Dimensionality reduction

In order to perform dimensionality reduction, we first identified 4000 genes that exhibited the most variable expression patterns across nuclei using the FindVariableFeatures function in Seurat with the mean.var.plot selection method, which aims to identify variable features while controlling for the strong relationship between variability and average expression, and otherwise default parameters. Gene-level data were then scaled using the ScaleData function in Seurat with default parameters. To examine dimensionality, we first performed a linear dimensional reduction using the RunPCA command with the maximum possible number of dimensions ("dim" set to 50). We then used Seurat's JackStraw, ScoreJackStraw, and JackStrawPlot functions to estimate and visualize the significance of the first 50 principal components (PCs), and the Elbow plot function to visualize the variance explained by the first 50 PCs. Because all 50 PCs were highly statistically significant, and no drop off was observed in variance explained across PCs, we used all 50 PCs for non-linear dimensional reduction (Uniform Manifold Approximation and Projection, UMAP) using the RunUMAP function in Seurat. For RunUMAP, "min.dist" was set to 0.5, "n.neighbors" was set to 50, and "metric" was set to "euclidean".

## Clustering

Prior to clustering, nuclei were embedded into a K-nearest neighbor (KNN) graph based on euclidean distance in UMAP space, with edge weights based on local Jaccard similarity, using the FindNeighbors function in Seurat (k.param = 50, dims = 1:2, prune.SNN = 0). Clustering was then performed using Seurat's FindClusters function using the Louvain algorithm with multilevel refinement (algorithm = 2). This final step was performed twice using two different resolution parameters to generate both coarse- and fine-grained structural descriptions of the underlying data, facilitating the investigation of both major cell types as well as smaller subpopulations. More coarse-grained clustering (resolution = 0.01) identified 15 1° clusters and fine-grained clustering (resolution = 1.3) identified 53 2° clusters.

## Cluster marker gene analysis

The biological identities of specific clusters were investigated using a multi-pronged approach that incorporated unbiased analysis of cluster-specific marker genes as well as a supervised examination of previously established marker genes. Cluster-specific marker genes were identified using the FindAllMarkers function in Seurat. Briefly, this function compares gene expression within each cluster to gene expression across all other clusters and calculates Bonferroni-adjusted *p* values using a Wilcoxon rank-sum test. Functional enrichment analysis of GO categories among cluster-specific marker genes was investigated by first converting cichlid gene names to their human orthologs and then performing functional enrichment analysis using ToppGene Suite with default parameters. Enrichment results that survived FDR-adjustment (*q* < 0.05) were considered statistically significant. Established cell type-specific and neuroanatomical marker genes were

identified from the literature (Supplementary Data File 2) and were intersected with the output from FindAllMarkers to generate further insight into the biological identity of clusters.

## Assignment of nuclei to test subjects

To match individual nuclei to individual test subjects, we used Demuxlet to match variants identified in snRNA-seq reads to variants identified from the genomic sequence of each subject[66]. First, genomic DNA from every test subject was collected and sequenced. In total, 276.7 Gbp of sequenced reads were assigned quality scores ≥30 (91.4% of all reads). The corresponding FASTQ files were filtered and aligned to the *Maylandia zebra* Lake Malawi cichlid genome assembly (NCBI RefSeq assembly accession: GCF_000238955.4, M_zebra_UMD2a). The resulting bam file was sorted, duplicates removed, read groups added, and indexed using Picard tools v2.23.3. Variants were then called using GATK v4.1.8.1 HaplotypeCaller using the M. zebra umd2a reference genome. For each sampling pool, individual vcf files were merged, resulting in ten files (one for each pool). These files were then filtered to retain only variants that varied among individuals within a pool. For each pool, only single nucleotide polymorphisms (SNPs) for which (1) at least one individual from the pool had a different genotype from the other individuals and (2) no individuals had missing data, were used as input to Demuxlet. The number of SNPs used ranged from 112,385 to 357,177, with a mean of 241,780 ± 22,369 per pool.

Next, variants were called from snRNA-seq reads following a similar pipeline. Reads from Cell Ranger's output bam file were filtered for those that passed the quality control metrics described above using samtools v1.11. The resulting bam file was sorted, duplicates removed, read groups added, and indexed using Picard tools. Variants were then called using GATK HaplotypeCaller using the M. zebra umd2a reference genome and without the MappingQualityAvailableReadFilter to retain reads that were confidently mapped by Cell Ranger (MAPQ score of 255). The SNPs from the snRNA-seq reads and the genomic DNA were used as input to Demuxlet, which computed a likelihood estimation that each nucleus belonged to each individual in the pool. Nuclei were assigned to the individual test subject with the greatest probability estimated by Demuxlet.

## Identification of IEG-like genes

Three canonical IEGs (*c-fos, egr1, npas4*) were used to identify additional genes exhibiting IEG-like expression across clusters. For each of these three IEGs, nuclei were split into IEG-positive versus IEG-negative nuclei within each of the 53 2° clusters. Within each cluster, differential gene expression was analyzed between IEG-positive versus IEG-negative nuclei using the FindMarkers function in Seurat, with "logfc.threshold" set to 0, and "min.pct" set to 1/57 (57 was selected as this was the number of nuclei in the smallest cluster). Within each cluster, any genes that did not meet this criterion were excluded and were assigned a *p* value of 1. Because FindMarkers requires at least three nuclei to be present in both comparison groups, clusters that contained less than three IEG-positive nuclei were excluded. Genes that were detected in the majority of clusters, and that were significantly (*p* < 0.05) upregulated in IEG-positive nuclei in the majority of those clusters were considered to be significantly co-expressed with each individual IEG. Genes that were significantly co-expressed with all three IEGs were used as IEG-like markers for downstream analyses of IEG-like expression.

## Differential IEG expression

Building-, quivering-, and gonadal-associated IEG expression was analyzed in 1° and 2° clusters, gene-defined populations within 1° and 2° clusters, and gene-defined populations regardless of cluster. To do this, we calculated an IEG score for each nucleus, equal to the number of unique IEG-like genes (*n* = 25) expressed. Building-, quivering-, and gonadal-associated differences in IEG score were analyzed using a beta-binomial model in which the number of IEG-like genes observed as well as the number of the IEG-like genes not observed were tracked as indicators of recent neuronal excitation. This analysis was performed using the 'PROreg' v1.2 package in R ("BBmm" function, m = 25)[67]. Because castle-building, quivering, and GSI were correlated with one another, we analyzed expression using a sequence of beta-binomial mixed-effects models in which different pairwise combinations of predictor variables (building, quivering, and GSI) competed to explain variance in IEG score. These models also included nested random terms to account for variance explained by individual variation, pair, pool, and RNA isolation/cDNA library generation batch. Within this framework, we ran the following seven models, which allowed building (analyzed as either a binary or a continuous variable), quivering, and GSI to compete in all possible combinations to explain variance in IEG score:

(1) IEG score ~ **BAI** + **log(quivering events)** + *(subject/pool/batch)* + *(subject/pair)*
(2) IEG score ~ **BAI** + **GSI** + *(subject/pool/batch)* + *(subject/pair)*
(3) IEG score ~ **BAI** + **log(quivering events)** + **GSI** + *(subject/pool/batch)* + *(subject/pair)*
(4) IEG score ~ **Condition** + **log(quivering events)** + *(subject/pool/batch)* + *(subject/pair)*
(5) IEG score ~ **Condition** + **GSI** + *(subject/pool/batch)* + *(subject/pair)*
(6) IEG score ~ **Condition** + **log(quivering events)** + **GSI** + *(subject/pool/batch)* + *(subject/pair)*
(7) IEG score ~ **log(quivering events)** + **GSI** + *(subject/pool/batch)* + *(subject/pair)*

We defined significant building-, quivering-, and gonadal-associated IEG effects as those in which (1) the raw *p* value for the corresponding fixed effect (for building, BAI and condition; for quivering, log-normalized quivering; for gonadal, GSI) was significant (*p* < 0.05) in every model and (2) the harmonic mean *p* value across models was significant after adjusting for multiple comparisons for all genes and populations analyzed (hmp_adj <0.05). To calculate the harmonic mean *p* value, we used the "harmonicmeanp" v3.0 package in R. Thus, building-associated IEG effects were considered significant if the raw *p* value for the effect of condition and BAI <0.05 in models 1–6, and if the harmonic mean *p* value across models 1–6 was significant after adjusting for multiple comparisons across all cell populations (5% false discovery rate).

## Building-, quivering-, and gonadal-associated gene expression

Building-, quivering-, and gonadal-associated gene expression was analyzed within 1° and 2° clusters using a multiple linear mixed-effects regression approach with the "glmmSeq" v0.5.5 package for R[68]. Because castle-building, quivering, and GSI were correlated with one another, we analyzed expression using a sequence of linear mixed-effects regression models in which different pairwise combinations of predictor variables (building, quivering, and GSI) competed to explain variance in gene expression. These models also included nested random terms to account for variance explained by individual variation, pair, sample pool, and 10X Chromium run. Thus, the sample size was equal to the number of individuals (*n* = 38), with many repeated observations being recorded from each individual (equal to the number of nuclei sampled from that individual as assigned to the cluster being analyzed). Building was analyzed both as a continuous variable (BAI) and as a binary categorical variable (behave versus control).

We defined build-DEGS, quiver-DEGs, and gonad-DEGs as genes (within clusters) in which expression was significantly (raw *p* < 0.05) associated with the corresponding fixed effect (for build-DEGs, BAI and condition; for quiver-DEGs, log-normalized quivering; for gonad-DEGs, GSI) in every model, and additionally in which the harmonic

mean $p$ value across models was significant after adjusting for multiple comparisons for all genes and all clusters (5% false discovery rate). For each model, the dispersion was estimated for each gene using the "DESeq2" v1.38.3 package for R[69], using parameters recommended for single-cell datasets (fitType = "glmGamPoi", minmu = 1e-06). Size factors for each gene were calculated using the "scran" v1.26.2 package in R[70], using default parameters, except that max.cluster.size was set to the number of nuclei assigned to the cluster being analyzed. Genes that were not observed in 19/19 pairs were excluded from analysis.

### Estrogen response element detection
Estrogen receptors are hormone-dependent transcription factors capable of regulating target gene expression by binding to specific DNA sequences called estrogen response elements. Estrogen response elements can be easily identified by their prototypic motif of AGGTCA separated by a 3-base spacer[71]. Genes within 25 kb of an estrogen response element motif were identified and the location of the estrogen response element relative to the gene was recorded as either intragenic (estrogen response element within the start site to the 3' polyA tail), promoter (estrogen response element ≤5 kb upstream of the gene), or distal (all other locations less than 25 kb away from the gene). We identified the gene closest to each estrogen response element using the closest command in bedtools v2.29.1.

### Building-, quivering-, and gonadal-associated proneurogenic gene expression
Building-, quivering-, and gonadal-associated proneurogenic gene expression was analyzed in 1° and 2° clusters, gene-defined populations within 1° and 2° clusters, and gene-defined populations regardless of cluster using the same general approach described for IEG expression, except that building-associated effects were defined as those that were significantly associated with a condition (building or control) in all models. Because we did not expect neurogenesis or associated cellular processes to proceed over <100 min timescales, we did not additionally require effects to be significantly associated with BAI in all models.

### Building-associated changes in cell proportions
Behavior-associated differences in cell type-specific proportions were analyzed for 1° and 2° clusters with a binomial mixed-effects regression model using the "glmer" function within the "lme4" v1.1-31 package in R[72]. The model included condition, GSI, and quivering as fixed effects, and included a random term for individual variation. 1° cluster proportions were calculated as the proportion of all nuclei assigned to each 1° cluster, and 2° cluster proportions were calculated as the proportion of 1° parent cluster nuclei assigned to each 2° daughter cluster. Thus, each nucleus was treated as an observation with a binary outcome (either an instance of the target cluster or not) from an individual that could be explained by condition, quivering, or GSI. $p$ values were estimated using the 'lmerTest' v3.1-3 package in R[73], and qvalues were calculated using the "qvalue" v2.30.0 package in R[74]. Building-associated effects were defined as those that were significant after accounting for multiple comparisons across all clusters with a false discovery rate of 5% ($q < 0.05$).

### Cluster-specific enrichment of gene sets
To test if genes associated with the evolution of bower construction behavior (CDGs and CDG module genes identified through comparative genomics) were enriched in specific cell populations, we first calculated a gene set score for each nucleus, equal to the total number of unique behavioral evolution genes expressed. Because the gene set score could be impacted by the total volume of sequence data sampled from each nucleus, we divided the gene set score by the total number of unique genes expressed in each nucleus. To quantify enrichment, a $Z$-test was then used to compare normalized gene set scores for all nuclei within a cluster compared to all other nuclei. The distribution of the normalized values was assumed to be normal according to the central limit theorem and population standard deviation was approximated using sample standard deviation.

Secondly, the effect size, as measured by Cohen's d, of the results were compared to those of random gene lists. To prevent differences in the overall amount of expression between random genes and genes of interest from skewing results, we identified random genes that had approximately equal number of UMIs expressed as a whole compared to the genes of interest. This was achieved by first ranking all the genes from the highest number of UMIs expressed to the lowest. Next, for each gene of interest, we generated a pool of 100 random genes that (1) ranked most closely to the gene of interest and (2) were not genes of interest themselves. Lastly, random gene lists were created by choosing one gene at random from each pool. The enrichment test described above compared effect sizes for the genes of interest versus effect sizes for 10,000 random gene lists. Clusters that were (1) significantly enriched compared to other nuclei according to the process above and (2) had greater effect sizes for the genes of interest than 95% of randomly permuted gene lists were considered to be significantly enriched.

### RG subclustering
RG subclusters were determined using the same general procedure used for clustering 1° and 2° clusters but restricted to only those nuclei assigned to 1.1_RG and 1.2_RG.

### Analysis of castle-associated genomic divergence
In order to identify potential behavior-associated genomic variants, comparative genomic analyses were performed using genomic sequence data collected from 27 Lake Malawi cichlid species[75]. Fixation indices ($F_{ST}$) were calculated for polymorphic variants in two separate analyses using vcftools v0.1.17. The first analysis compared pit-digging ($N = 11$) versus castle-building ($N = 9$) species, and the second compared rock-dwelling ($N = 7$) versus castle-building ($N = 9$) species. Variants for which sequence data was missing from 50% or more of species in either group were excluded from analysis. $F_{ST}$ analyses were performed separately using the --weir-fst-pop and --fst-window-size 10000 flag to calculate $F_{ST}$ across 10 kb bins in vcftools. Then, bins where $F_{ST}$ was greater than 0.20 in the pit-castle comparison and 0.20 in the rock-castle comparison were kept. Only 8.9% of FDR-adjusted significant ($hmp_{adj} < 0.05$) bins met these thresholds, indicating that the selected bins were extremely divergent between castle-building and non-castle-building species. Finally, genes located within 25 kb of bins that passed these thresholds were identified using bcftools v1.11 with the closest command and the *M. zebra* genome as ref. [20]. Genes within 25 kb of highly divergent pit-castle and rock-castle bins are referred to here as CDGs.

### CDG co-expression and module analysis
Modules of co-expressed CDGs were analyzed using weighted gene correlation network analysis (WGCNA) using the "WGCNA" v1.70-3 package in R. CDGs that were not observed in any nucleus were excluded from the analysis. The normalized gene expression data for CDGs was used as the input gene expression matrix and the function pickSoftThreshold was used to determine the optimal soft-thresholding power. We determined the optimal soft-thresholding power to be 1 because it was the lowest power for which the scale-free topology fit index reached 0.90. Then an adjacency matrix was created from the input gene expression matrix using the adjacency function with power = 1, type = "signed" and otherwise default parameters. The adjacency matrix was used as the topological overlap matrix (TOM) and the dissimilarity matrix was calculated as 1 − TOM. To detect modules, k-means clustering was performed using all possible values of k and the results were compared to determine the optimal k. First, a

distance matrix was constructed from the dissimilarity matrix produced by WGNCA using the dist function in R. Next, the function pam from the R package "cluster" v2.1.0 was used to cluster the distance matrix with diss = T, otherwise default parameters, and k set to the value that produced the highest average silhouette width for all genes. Briefly, silhouette width is a measure of the similarity of an element (here a gene) to its own cluster (here a gene module) versus other clusters. We found that k = 2, resulted in the greatest average silhouette width. The strength of the CDG module was evaluated using a two-sampled Welch t-test comparing the silhouette width and gene-gene correlations for CDGs within the CDG module versus CDGs outside the CDG module. To analyze the relationship between the CDG module and signatures of RG quiescence, the correlation coefficient was calculated based on the number of genes in the CDG module expressed in each nucleus versus the number of quiescent markers expressed in each nucleus. We compared the correlation coefficient against a permuted null distribution that was generated by randomly shuffling the expression values of each gene in the CDG module across nuclei 10,000 times.

### Spatial transcriptomic pre-processing and quality control

Base Call files were demultiplexed into FASTQ files and processed with Space Ranger v1.3.1 (10X Genomics). Reads were aligned to the *M. zebra* umd2a reference assembly as described above for snRNA-seq. Following these steps, Space Ranger generated three filtered feature-barcode matrices containing expression data for a total of 32,471 features (corresponding to annotated genes). Spots with 0 UMIs were removed, resulting in 6707 spots used in downstream analysis.

### Spatial integration of snRNA-seq clusters

To predict locations of specific snRNA-seq identified clusters in spatial transcriptomics data, an "anchor"-based integration workflow in Seurat was used. First, both the snRNA-seq and spatial data were normalized using the SCTransform function in Seurat. Next, anchors were identified between the reference snRNA-seq and the query spatial data using FindTransferAnchors in Seurat, and a matrix of prediction scores was generated for each cluster in every spot using the TransferData function in Seurat. The maximum prediction score across clusters was not uniform; therefore, we normalized the values between 0 and 1 in order to enable meaningful comparisons across cell types.

### Cell-cell communication analysis

To assess potential directional communication between cell populations, cell–cell communication analysis was performed using the R package CellChat v1.5.0. Briefly, CellChat estimates the strength of the potential ommunication between populations (measured as "connection weight") from a gene expression matrix based on a database of human ligand-receptor interactions. In order to find the connection weights between 1° and 2° clusters, two copies of the gene expression matrix were generated, and the cells in the first copy were assigned 1° cluster labels and the cells in the second copy were assigned 2° cluster labels. We also sought to analyze connection weights among additional gene-defined populations that demonstrated behavior-associated IEG expression. To achieve this, the gene expression matrices for cells from these populations were duplicated again. Before connection strengths were evaluated, the human orthologs of the *M.zebra* gene names in the gene expression matrix were found. Since the gene expression matrix does not allow for duplicate gene names, e.g., many-to-one orthologs, values for the many-to-one ortholog with the greatest number of normalized counts were kept and others were excluded from analysis. Next, a CellChat object was created using the createCellChat function. Over-expressed genes and over-expressed interactions were found using the identifyOverExpressedGenes and identifyOverExpressedInteractions functions, respectively. Next, connection weights were calculated using the

computeCommunProb function with the method for computing the average gene expression per cell group set to truncatedMean, trim set to 0.1, and population.size set to FALSE. Then, the cellular communication network was inferred and aggregated using the filterCommunication and aggregateNet functions. The receptor-ligand and the signaling pathway weights were saved using subsetCommunication with the slot.name parameter set to "net" and netP respectively.

### Mediation analyses

Mediation analysis tests if the relationship between a predictor and outcome variable is influenced by possible mediator variables. Analyses of quivering and GSI as possible mediators of bower activity were performed using the R package BruceR v0.8.9 (https://psychbruce.github.io/bruceR/). In these analyses, the group was the predictor (categorical, build or control), bower activity index was the continuous outcome, and quivering (log-normalized) and GSI were investigated as possible independent mediators or as possible serial mediators (in both possible orders), in all cases with nsim set to 1000. For all downstream analyses, which included larger sets of multiple possible mediators, multivariate outcomes in some cases, and missing observations for some individuals in some cases ($RG_8$ and $RG_1$ nuclei were only sampled from 32/38 and 37/38 males, respectively), we performed regularized multiple mediation analysis using the R package mmabig v3.1.0[76–78]. Briefly, this analysis uses a regularization approach to identify candidate mediators and to estimate their collective and individual indirect effects on the relationship between the predictor variable (i.e., group) and outcome (i.e., building activity, rebalancing, build-IEG+ score, and RG biology). We used this approach to investigate possible upstream and downstream mediation effects for building activity, build-IEG+ excitation, neuronal rebalancing, RG biology, and RG CDG expression. Candidate mediators were identified using the data.org.big function with alpha set to 0.2 (where 1 indicates a LASSO penalization and 0 indicates a ridge penalization), alpha1 set to 0.05, alpha2 set to 0.05, and lambda set to "exp(seq(log(0.001), log(5), length.out = 10,000))". Estimation of indirect effects was performed using the mma.big function with alpha set to 0.2, alpha1 set to 0.05, alpha2 set to 0.05, n2 set to 1000, and lambda set to "exp(seq(log(0.001), log(5), length.out = 1000))".

### Reporting summary

Further information on research design is available in the Nature Portfolio Reporting Summary linked to this article.

## Data availability

The snRNA-seq and spatial transcriptomics data generated in this study are deposited and publicly available in the National Center for Biotechnology Information (NCBI) Gene Expression Omnibus (GEO) under accession code GSE217619 (spatial transcriptomics data are deposited as a SubSeries). The DNA data generated in this study are deposited and publicly available in the NCBI BioProject databank under accession code PRJNA867404. The reference genome used in this study was the *Maylandia zebra* UMD2a RefSeq assembly, deposited and publicly available in the NCBI BioProject databank under accession code GCF_000238955.4. Source data are provided with this paper.

## Code availability

Code use for core analyses is publicly available at https://github.com/streelmanlab/cichlid_sn/, https://doi.org/10.5281/zenodo.8030021.

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

## Acknowledgements

We thank our collaborators Ashley Parker and Drs. Swantje Graetsch, Manuel Stemmer, and Herwig Baier for valuable feedback during the early stages of the project; Dr. Nicholas Johnson for suggestions regarding statistical analysis of IEG co-expression; Dr. Justin Rhodes for insightful feedback on IEG expression analysis; Dr. Cori Bargmann for thoughtful feedback on early drafts of the manuscript; Cristina Baker for her critical role in the initial development of spatial transcriptomics wetlab pipelines; and the Georgia Tech Petit Institute Genome Analysis and Molecular Evolution Cores for their integral roles in sample processing and sequencing, respectively. This work was supported in part by NIH R01GM101095 and R01GM144560 to J.T.S., NIH F32GM128346 to Z.V.J., NIH R35 GM139594 to P.T.M., NSF Graduate Research Fellowship DGE-2039655 to T.J.L., and Human Frontiers Science Program RGP0052/2019 to J.T.S.

## Author contributions

General: Z.V.J. initially conceived of the experiment and Z.V.J., J.T.S., and B.E.H. developed and designed it. Z.V.J. and B.E.H. performed all wetlab work (see details below under "Wetlab"). T.J.L. pre-processed behavioral and depth data, including in part spatial and temporal registration of both data streams and temporal anchoring to experimental endpoints. G.W.G. pre-processed snRNA-seq, DNA-seq, and spatial transcriptomics data. Z.V.J. and G.W.G. performed downstream data analysis (see details below under "Drylab"). B.E.H. matched snRNA-seq data to published neuroanatomical expression profiles (see details below under "Drylab"). Z.V.J. took the lead on writing the manuscript with critical feedback from G.W.G., J.T.S., B.E.H., and P.T.M. Z.V.J. took the lead on designing and creating figures with contributions from B.E.H., G.W.G., and T.J.L., and with critical feedback from G.W.G., J.T.S., B.E.H., P.T.M., and T.J.L. J.T.S. mentored and funded Z.V.J., B.E.H., and G.W.G., and P.T.M mentored and funded T.J.L. on the project. J.T.S. funded snRNA-seq, DNA-seq, and spatial transcriptomics experiments. Wetlab: Z.V.J. and B.E.H. developed and optimized a single nucleus isolation protocol for cichlid telencephala. Z.V.J. and B.E.H. performed all behavioral assays, surgeries, and downstream nuclei isolations for snRNA-seq. Z.V.J. performed DNA isolations for matching nuclei to subjects. B.E.H. performed all behavior assays for spatial transcriptomics. Z.V.J. and B.E.H. performed surgeries for spatial transcriptomics. B.E.H. performed all downstream wetlab work for spatial transcriptomics. The Petit Institute Genome Analysis Core at GT performed library preparation for snRNA-seq, DNA-seq, and

spatial transcriptomics. The Petit Institute Molecular Evolution Core at GT performed sequencing for snRNA-seq, DNA-seq, and spatial transcriptomics. Drylab: Z.V.J. performed clustering and cluster marker analysis. B.E.H. systematically surveyed the literature to determine conserved neuroanatomical expression patterns of ligand, receptor, nTF, and other cell type-specific marker genes in the teleost telencephalon. B.E.H., G.W.G., and Z.V.J. collaboratively identified markers of radial glial quiescence, cycling, and neuronal differentiation. Z.V.J. and G.W.G. collaboratively developed many analytical approaches. Z.V.J. conducted behavioral, IEG co-expression, IEG, DEG, pNG, cell proportion, and gene set enrichment (for biological categories) analyses. G.W.G. matched nuclei to subjects and conducted comparative genomics, gene orthologue calling, ERE detection, gene module detection, and cluster enrichment (for gene lists) analyses. G.W.G. performed spatial integration of clusters and B.E.H. matched spatial transcriptomic profiles to brain regions. Z.V.J. and G.W.G. performed cell–cell communication analyses.

## Competing interests

The authors declare no competing interests.
