## [Peer Review File · Nature Communications]

The correct author order is provided in the manuscript file.
Please note that there are three co-first authors in total.REVIEWER COMMENTS

Reviewer #1 (Remarks to the Author):

General comments:

This study by Johnson et al brings together behavior, transcriptomics, and comparative genomics to find transcriptomic profiles, cell types, and selective changes in genomic regions associated with differences in bower building behavior in cichlid species. An intriguing find is that changes in glial cells at the surface of the pallium that eventually migrate into the hippocampus homolog appear to be a driving cell types for bower building differences between species. Overall, the study is an impressive tour de force effort. At the same time, there are number of issues with the study that need cleaning up.

The results section of the supplementary document is like another paper, or a much longer version of the main text results. This uses up reader energy. I think the author need to cut out the redundancy in the supplementary results, and really make it supplementary to the main text and not another expanded version of the main text.

The abstract mentions comparative genomics across 27 species to trace bower-genome associated evolution. I can't find anywhere in the paper the mention or list of those 27 species, their genome accession numbers, or otherwise. I assume that this might have something to do with the gene divergence analyses in Figure 5A, but that is hard to tell.

Although the paper has a lot of text and figures, it is not always easy to follow. This is because not all axis are labelled, there are many abbreviations, and not all ideas or sections of the paper well connected.

The title implies that the study is mainly about glia cells in social behavior. But the paper is more than that. It is about the transcriptome and genetic changes in neuron and glia brain cells, involved in building behavior for a social reward. How about.

“Transcriptome cellular profiling reveals roles for both neuron and glia in a socially-motivated building behavior”

Specific comments:

The immediate early gene section starting at line 103 is not clear about what was done to the animals before sacrifice. The supplemental methods section is also not clear. It looks like the authors are trying to measure activity-dependent induction in the brains of animals that build bowers and controls without building, using single cell transcriptomics. It looks like it might be a sufficient or good design, but not clear. Did they follow some of the protocols of Burmeister et al 2005 in PLoS Biology, the first study I believe of behaviorally-driven IEG in cichlids, if not fish. There they had the fish in tanks overnight, turned the lights on, and then measured behavior activity and thereafter IEG activity in the brain.

In the current study, where the males carried to the behavioral tank right before the procedure started, or the night before. If right before, then there could be stressed induced IEG being measured as well. How long prior to behavioral test did the animals have a pre-training session? This could make a difference. In the tissue sampling methods, it says the brains were collected 11am-2pm EST (3-5 h after the lights come on). Later in the behavioral analyses, there is mention a 90 min window in which behavior was analyzed. I think the authors need to include a timeline protocol figure, of what was done in the hours to day before the fish were sacrificed and brains removed. If there are any confounding variables, this needs to be considered as well.

Figure 5A is impressive, in terms of the genomic block showing a consistently high divergence, including within and between genes. This makes me wonder if the author really are looking at a homologous genome region across species, or at paralogous regions? Can this be double checked

Line 301. Need to make clearer in a sentence or two what cell-cell communication analyses is; grammatically, one could interpret this as a neurophysiology interaction between two neurons. But the authors I presume are refer to gene expression analyses between two cells?

Figure 2D. Why is the x axis time course going in the reverse direction, from 90 to 30 min?

Figure 3D. What is the y axis variable? Needs an axis label.

Reviewer #2 (Remarks to the Author):

Inspired by previous work in inbred mice, where single nucleus/single cell RNA sequencing had been used to profile neuronal signatures of behaviour, the authors of the present study apply the same strategy to bower-building versus non-bower-building males of cichlid fishes from Lake Malawi (N=19 each). This was possible because of the extensive previous work by the main PI and his team in this model system and on this behavioural phenotype. The authors present an impressive amount of data and experimental results that indicate a link between bower building and a number of genes previously associated with behavioural phenotypes in cichlids, ultimately suggesting that CDG regulation in glia cells is at the core of the evolutionary transitions towards bower building in cichlids.

Overall, this integrative study clearly ranks among the most thorough examinations of a behavioural phenotype in cichlids at the cellular level, and although no causal relationships could be established, the results presented are "plausible" at various levels. For example, the genes (and genetic networks) identified make a lot of sense in the light of previous results, and such does the link to the estrogen signalling cascade. Also, the putative link to the previously known 19 Mbp inversion appears a highly promising target for future research. This is why I think that this work should be published in a journal such as Nature Communications.

At the same time, the wealth of analyses and data presented make this manuscript rather technical and in parts difficult to read and follow. Likewise, although I find that the authors did a great job in packing such an amount of experimental work and results into their manuscript, I feel that some of the figures are overcrowded, in particular figures 1, 4 and 5, and I wonder whether it wouldn't make sense to move some of the display items into the supplement.

Finally, I would have hoped for a bit more "biology" in the introduction section, in particular about bower building in cichlid (and other) fishes, plus - ideally - a photograph of a bower built by a Mchenga conophoros male from the wild (or at least from the lab) as display item in Fig. 1, which clearly would help those readers that have not yet been diving or snorkeling in the Great Lakes of Africa to get an impression of what a bower looks like. With respect to "biology", it could be mentioned that bower building has evolved multiple times not only in Lake Malawi cichlids but also in other cichlid assemblages/lineages, and that this behaviour is also found in a number of other clades of fish. Perhaps it can also be made more explicit that females do not show this behavioural phenotype. In this latter respect, I wonder if the authors have ever considered to also include the cellular profiling of female brains (e.g., to test if these are, at least in parts, similar to non-bower-building males or if the differences between "behaving" and "non-behaving" males occurs within a male-specific cell population)? Given that *cyp19a1* appears to be differentially expressed, a potential link to sex determination/differentiation should/could be investigated in the future.

Minor issues:

- Abstract, "the putative fish hippocampus": maybe better say "putative hippocampal homologue" as in the main text
- line 27: maybe better "non-traditional model systems"?
- line 34: is it really the brain that is behaving, or wouldn't the brain need the entire organism around it to perform any behavioural task?
- line 38: should be "including cell populations in..."
- line 40: not sure if "bower behaviour" is the correct thing to say here, it should be "bower-building behaviour"
- line 45: are these really "castles"? I think that "bower" should do it here, too.
- line 238: "19 Mbp" instead of "19Mbp"
- line 340: should also be "cell populations".
- lines 381-385: I do not get this sentence.
- 396: "cis" in italics as elsewhere.

March 29, 2023

Dear Referees,

Thank you for your constructive feedback. We have substantially changed our manuscript in response to your concerns. In the few instances in which no changes were made in response to a point, we detail our reasoning why. Overall, we feel that your suggestions have resulted in a substantially improved manuscript, and hope that you find it suitable for publication. Below we provide a copy of your comments in full (bold) with detailed point-by-point responses to major and minor concerns (numbered points).

Sincerely,

Jeffrey T. Strelman
Professor, Chair

Patrick T. McGrath
Associate Professor

Zachary V. Johnson
Postdoctoral Fellow

Point-by-point responses to reviewers:

REVIEWER 1:

General comments:

This study by Johnson et al brings together behavior, transcriptomics, and comparative genomics to find transcriptomic profiles, cell types, and selective changes in genomic regions associated with differences in bower building behavior in cichlid species. An intriguing find is that changes in glial cells at the surface of the pallium that eventually migrate into the hippocampus homolog appear to be a driving cell types for bower building differences between species. Overall, the study is an impressive tour de force effort. At the same time, there are number of issues with the study that need cleaning up.

1. The results section of the supplementary document is like another paper, or a much longer version of the main text results. This uses up reader energy. I think the author need to cut out the redundancy in the supplementary results, and really make it supplementary to the main text and not another expanded version of the main text.

Response: We agree with this comment and have carefully gone through the supplementary text to identify and cut redundancies. In total, we cut more than a page of text from the Supplementary Results and streamlined what we felt was essential.

2. The abstract mentions comparative genomics across 27 species to trace bower-

genome associated evolution. I can't find anywhere in the paper the mention or list of those 27 species, their genome accession numbers, or otherwise. I assume that this might have something to do with the gene divergence analyses in Figure 5A, but that is hard to tell.

Response: Thank you for identifying this oversight. We have added a Supplementary Table (S14) with species names, phenotypes, and accession numbers.

3. Although the paper has a lot of text and figures, it is not always easy to follow. This is because not all axis are labelled, there are many abbreviations, and not all ideas or sections of the paper well connected.

Response: We appreciate this feedback. We identified three panels (Fig. 4F; Fig. 6B, top; Fig. 7E). We added a y-axis label to Fig. 4F, and for space considerations we have added clear descriptions of the x-axis for Fig. 6B and Fig. 7E to the figure legends. We also agree that unnecessary abbreviations make papers more difficult to follow. We have removed abbreviations for estrogen response elements (originally EREs) and radial glia (originally RG). We take the final comment very seriously and have attempted to address this concern by adding and rephrasing text throughout the paper, with special focus on 1) improving transitions, 2) strengthening justifications for specific analyses, and 3) clarifying interpretations of results and corresponding hypothesized models.

4. The title implies that the study is mainly about glia cells in social behavior. But the paper is more than that. It is about the transcriptome and genetic changes in neuron and glia brain cells, involved in building behavior for a social reward. How about. "Transcriptome cellular profiling reveals roles for both neuron and glia in a socially-motivated building behavior"

Response: We agree the original title did not sufficiently reflect the breadth of results. We have revised the title of the manuscript to: "Cellular profiling of a recently-evolved social behavior." We considered expanding the title to include "neurons and glia" in the title, but were concerned that it sounded overly general and may leave a potential reader with the incorrect impression that "everything is involved in a recently-evolved social behavior".

Specific comments:

5. The immediate early gene section starting at line 103 is not clear about what was done to the animals before sacrifice. The supplemental methods section is also not clear. It looks like the authors are trying to measure activity-dependent induction in the brains of animals that build bowers and controls without building, using single cell transcriptomics. It looks like it might be a sufficient or good design, but not clear. Did they follow some of the protocols of Burmeister et al 2005 in PLoS Biology, the first study I believe of behaviorally-driven IEG in cichlids, if not fish. There they had the fish in tanks overnight, turned the lights on, and then measured behavior activity and thereafter IEG activity in the brain.

In the current study, where the males carried to the behavioral tank right before the procedure started, or the night before. If right before, then there could be stressed induced IEG being measured as well. How long prior to behavioral test did the animals have a pre-training session? This could make a difference. In the tissue sampling methods, it says the brains were collected 11am-2pm EST (3-5 h after the lights come on). Later in the behavioral analyses, there is mention a 90 min window in which behavior

was analyzed. I think the authors need to include a timeline protocol figure, of what was done in the hours to day before the fish were sacrificed and brains removed. If there are any confounding variables, this needs to be considered as well.

Response: Males were not introduced to tanks right before, so stress caused by transfer and introduction to a new tank was not a confounding variable. Introduction to a behavior tank initiated a pre-trial period which varied in length (at least a day) depending on when a given male initiated building. The sand was then smoothed immediately before lights off, behavioral trials were initiated, and males were not collected until at least the following day. This design controlled for the potentially confounding effect of novelty (i.e. the first time building, exposure to novel individuals, or exposure to a novel environment) on brain gene expression. Males were collected between 11-2pm ET, and behavior in the 100-minute period prior to collection was used for analysis of IEG expression. We have made a number of changes to make this design clearer in the main text (Line 122-123) and supplement (Lines 597-614).

6. Figure 5A is impressive, in terms of the genomic block showing a consistently high divergence, including within and between genes. This makes me wonder if the author really are looking at a homologous genome region across species, or at paralogous regions? Can this be double checked

Response: Our previous data and the data from other organisms such as the ruff, salmon, butterflies, and various plants suggest this pattern is most likely a genome rearrangement (e.g. an inversion). We are actively pursuing independent means of investigating this region, and our findings thus far are consistent with this interpretation.

7. Line 301. Need to make clearer in a sentence or two what cell-cell communication analyses is; grammatically, one could interpret this as a neurophysiology interaction between two neurons. But the authors I presume are refer to gene expression analyses between two cells?

Response: We agree this point needs to be clear and have added additional information to the main text description (Lines 382-386).

8. Figure 2D. Why is the x axis time course going in the reverse direction, from 90 to 30 min?

Response: Each point on the x-axis represents temporally binned behavior centered on that timepoint. The values are in “reverse” order because it is measured in minutes *before* flash freezing of the brain. We feel this is appropriate because IEG expression time courses are typically discussed in terms of “minutes following a stimulus or behavior” and in the context of our experimental design, this matches that same framework, i.e. 60 minutes on the x-axis represents behavior performed 60 minutes in the past. We have added these details to the figure legend for clarification.

9. Figure 3D. What is the y axis variable? Needs an axis label.

Response: Thank you for catching this. Fixed!

REVIEWER 2:

Inspired by previous work in inbred mice, where single nucleus/single cell RNA sequencing had been used to profile neuronal signatures of behaviour, the authors of the present study apply the same strategy to bower-building versus non-bower-building males of cichlid fishes from Lake Malawi (N=19 each). This was possible because of the extensive previous work by the main PI and his team in this model system and on this behavioural phenotype. The authors present an impressive amount of data and experimental results that indicate a link between bower building and a number of genes previously associated with behavioural phenotypes in cichlids, ultimately suggesting that CDG regulation in glia cells is at the core of the evolutionary transitions towards bower building in cichlids.

Overall, this integrative study clearly ranks among the most thorough examinations of a behavioural phenotype in cichlids at the cellular level, and although no causal relationships could be established, the results presented are "plausible" at various levels. For example, the genes (and genetic networks) identified make a lot of sense in the light of previous results, and such does the link to the estrogen signalling cascade. Also, the putative link to the previously known 19 Mbp inversion appears a highly promising target for future research. This is why I think that this work should be published in a journal such as Nature Communications.

At the same time, the wealth of analyses and data presented make this manuscript rather technical and in parts difficult to read and follow. Likewise, although I find that the authors did a great job in packing such an amount of experimental work and results into their manuscript,

1. I feel that some of the figures are overcrowded, in particular figures 1, 4 and 5, and I wonder whether it wouldn't make sense to move some of the display items into the supplement.

Response: We appreciate this feedback and have modified Figures 1, 4, and 5 from the original submission by 1) splitting Figure 1 into two smaller figures (now Figures 1 and 2), one focused on behavior (with photos of bower in the wild and laboratory added) and the other focused on clusters; 2) splitting Figure 4 into two smaller figures (now Figures 5 and 6), one focused on neuronal rebalancing (with one panel moved to supplement) and the other focused on radial glial biology; and 3) reworking Figure 5 (now Figure 7) by relocating the CDG module co-expression network diagram (originally panel C) to the supplement and reorganizing the remaining panels. We feel that these changes have helped make the manuscript more accessible.

2. Finally, I would have hoped for a bit more "biology" in the introduction section, in particular about bower building in cichlid (and other) fishes, plus - ideally - a photograph of a bower built by a Mchenga conophoros male from the wild (or at least from the lab) as display item in Fig. 1, which clearly would help those readers that have not yet been diving or snorkeling in the Great Lakes of Africa to get an impression of what a bower looks like. With respect to "biology", it could be mentioned that bower building has evolved multiple times not only in Lake Malawi cichlids but also in other cichlid assemblages/lineages, and that this behaviour is also found in a number of other clades of fish. Perhaps it can also be made more explicit that females do not show this behavioural phenotype. In this latter respect, I wonder if the authors have ever considered to also include the cellular profiling of female brains (e.g., to test if these are, at least in parts, similar to non-bower-building males or if the differences between "behaving" and "non-behaving" males occurs within a male-specific cell population)?

Given that cyp19a1 appears to be differentially expressed, a potential link to sex determination/differentiation should/could be investigated in the future.

Response: We have added additional information about the biology of this behavior to the Introduction, including additional male-specific dimensions of the behavior (Lines 36-43). Additionally, we have added photographs of *Mchenga conophoros* above castle bowers both in the wild and in the lab (Fig. 1A-B). We have failed to find a place for additional descriptions of bower-building in other species (e.g. in Tanganyikan cichlids) without disrupting the flow of the paper, but we are open to further suggestions. We have not yet profiled female brains, and agree it would be an interesting line of future research, although it is beyond the scope of this study.

Minor issues:

3. Abstract, "the putative fish hippocampus": maybe better say "putative hippocampal homologue" as in the main text

Response: We agree this is more precise and have made the change.

4. line 27: maybe better "non-traditional model systems"?

Response: We have made this change (Line 27)

5. line 34: is it really the brain that is behaving, or wouldn't the brain need the entire organism around it to perform any behavioural task?

Response: We agree this is more precise and have corrected the language accordingly (Lines 29-31)

6. line 38: should be "including cell populations in..."

Response: We have made this change (Line 35)

7. line 40: not sure if "bower behaviour" is the correct thing to say here, it should be "bower-building behaviour"

Response: We agree and have made this change (Lines 32,37)

8. line 45: are these really "castles"? I think that "bower" should do it here, too.

Response: We have maintained this language based on previous work. Our concern with using "bower" here is that it encompasses both pit "craters" and castle "mounds", and these distinctions are important later in the paper (e.g. in comparative genomic analyses between pit versus castle species).

9. line 238: "19 Mbp" instead of "19Mbp"

Response: Thank you for catching this. Fixed (Line 281).

10. line 340: should also be "cell populations".

Response: Agreed, we have corrected this (Line 381-382).

11. lines 381-385: I do not get this sentence.

Response: We agree the original phrasing was confusing. The sentence was intended to explain our hypothesized behavioral circuit model based on the most strongly supported mediation relationships. We have re-written this sentence to make it clearer in Lines 414-417.

12. 396: "cis" in italics as elsewhere.

Response: Thank you for catching this, we have corrected (Line 487)

REVIEWERS' COMMENTS

Reviewer #2 (Remarks to the Author):

I think the authors have done a very good job in incorporating the feedback from the previous round of review. They have addressed my initial concerns. The manuscript (and figures) are more clear now.

There are three very minor comments left from my side:

L36: perhaps mention "behaviorally and eco-morphologically diverse"... species

L42: I guess "disease-associated genes" refers to humans? Please clarify

L305/306: maybe "add CDG score" before "was" to make this sentence more clear

Congratulations to this work,

Walter

Point-by-point responses to reviewers:

Reviewer 1:

I think the authors have done a very good job in incorporating the feedback from the previous round of review. They have addressed my initial concerns. The manuscript (and figures) are more clear now. There are three very minor comments left from my side:

Point 1: L36: perhaps mention "behaviorally and eco-morphologically diverse"... species

Response: We have added this exact language and agree it offers a more complete biological picture of cichlids and their significance (Line 35 in updated manuscript)

Point 2: L42: I guess "disease-associated genes" refers to humans? Please clarify

Response: We have added "human" before "disease-associated" and agree this avoids unnecessary confusion (Line 44 in updated manuscript)

Point 3: L305/306: maybe "add CDG score" before "was" to make this sentence more clear

Response: We have added "CDG module score" before "was" in both clauses and agree this improves clarity (Lines 250-251 in updated manuscript)